# Soil-borne fungi alter the apoplastic purinergic signaling in plants by deregulating the homeostasis of extracellular ATP and its metabolite adenosine

Christopher Kesten[1,2†‡], Valentin Leitner[1†§], Susanne Dora[1], James W Sims[3], Julian Dindas[4‡], Cyril Zipfel[4], Consuelo M De Moraes[3], Clara Sanchez-Rodriguez[1,5]*

[1]Department of Biology and Zürich-Basel Plant Science Center, Zürich, Switzerland; [2]Department for Plant and Environmental Sciences, University of Copenhagen, Copenhagen, Denmark; [3]Department of Environmental Systems Science, ETH Zürich, Zurich, Switzerland; [4]Institute of Plant and Microbial Biology and Zürich-Basel Plant Science Center, University of Zürich, Zürich, Switzerland; [5]Centro de Biotecnología y Genómica de Plantas, Universidad Politécnica de Madrid (UPM) – Instituto Nacional de Investigación y Tecnología Agraria y Alimentaria (INIA/CSIC), Pozuelo de Alarcón, Spain

**\*For correspondence:**
clara.sanchez@csic.es

[†]These authors contributed equally to this work

**Present address:** [‡]Lonza AG, Visp, Switzerland; [§]Institute of Science and Technology Austria (ISTA), Klosterneuburg, Austria

**Competing interest:** The authors declare that no competing interests exist.

**Abstract** Purinergic signaling activated by extracellular nucleotides and their derivative nucleosides trigger sophisticated signaling networks. The outcome of these pathways determine the capacity of the organism to survive under challenging conditions. Both extracellular ATP (eATP) and Adenosine (eAdo) act as primary messengers in mammals, essential for immunosuppressive responses. Despite the clear role of eATP as a plant damage-associated molecular pattern, the function of its nucleoside, eAdo, and of the eAdo/eATP balance in plant stress response remain to be fully elucidated. This is particularly relevant in the context of plant-microbe interaction, where the intruder manipulates the extracellular matrix. Here, we identify Ado as a main molecule secreted by the vascular fungus *Fusarium oxysporum*. We show that eAdo modulates the plant's susceptibility to fungal colonization by altering the eATP-mediated apoplastic pH homeostasis, an essential physiological player during the infection of this pathogen. Our work indicates that plant pathogens actively imbalance the apoplastic eAdo/eATP levels as a virulence mechanism.

## Editor's evaluation

This important paper reports how a fungal pathogen utilizes adenosine to perturb plant disease resistance and immune signaling. The authors convincingly show a key role of Ado/eATP in the alteration of apoplastic pH and pathogenesis. The research presented provides a foundation for the study of extracellular adenosine on purinergic signaling during plant-pathogen interactions.

## Introduction

Adenosine-5'-triphosphate (ATP) constitutes the energy currency of all living organisms and is the driving force of many cellular processes. In addition, it fulfills a broad range of tasks in signaling

mechanisms once it leaves the cytosol and becomes extracellular (eATP). In plants, eATP contributes to root hair growth, gravitropism, cell death, and to response to abiotic and biotic stresses (*Roux and Steinebrunner, 2007*; *Chen et al., 2017*; *Tanaka et al., 2014*; *Choi et al., 2014a*). ATP reaches the apoplast through transporters (*Thomas et al., 2000*; *Rieder and Neuhaus, 2011*) and secretory vesicles (*Kim et al., 2006*). Since eATP is involved in a broad selection of signaling processes, tight controlling mechanisms are required to regulate its concentration. These comprise apoplast facing apyrases and purple acid phosphatases that hydrolyze eATP to adenosine monophosphate (AMP; *Zrenner et al., 2006*; *Deng et al., 2015*). AMP is subsequently hydrolyzed by 5' nucleotidases (5'NT) to extracellular adenosine (eAdo; *Zrenner et al., 2006*) that is either taken up into to cytoplasm by the *Equilibrative nucleoside transporter 3* (ENT3; *Traub et al., 2007*), or further processed by the extracellular protein *Nucleoside hydrolase 3* (NSH3; *Jung et al., 2011*). NSH3 removes the sugar moiety of eAdo and generates adenine (Ade), which is transported into the cytoplast by a purine permease transporter (PUP; *Möhlmann et al., 2014*).

Mechanical wounding of the plasma membrane leads to a high release of ATP to the apoplast (*Tanaka et al., 2014*), increasing eATP concentration up to 80 nM (*Dark et al., 2011*), which is sufficient to activate the purinoreceptor *Does not respond to nucleotides* 1 (P2K1/DORN1) also known as the LecRK-I.9 (lectin receptor kinase I.9; $K_d$ ~46 nM; *Choi et al., 2014a*). Perception of eATP by DORN1/LecRK-I.9 induces cellular responses including increase of cytoplasmic $Ca^{2+}$ concentrations and of reactive oxygen species (ROS), activation of MAPK cascades by phosphorylation, and transcriptional reprogramming (*Cao et al., 2014*). Indeed, 60% of genes differentially regulated after application of exogenous ATP are differentially expressed during wounding processes (*Choi et al., 2014a*). The role of eATP as a damage-associated molecular pattern (DAMP) is supported by the susceptibility of *dorn1* mutants to different pathogens and their lower response to a beneficial endophyte (*Balagué et al., 2017*; *Nizam et al., 2019*; *Jewell et al., 2022*).

In mammals, Ado is also recognized as a primary messenger, being a key signal of immunosuppressive responses after being perceived by G-protein-coupled receptors (*Antonioli et al., 2014*; *Antonioli et al., 2019*). In plants, *Arabidopsis ent3nsh3* double mutant, affected in the Ado/ATP ratio at the apoplast, showed increased susceptibility to the necrotroph ascomycete *Botrytis cinerea,* connected to an attenuation of the upregulation of defense-related genes (*Song et al., 2006*; *Daumann et al., 2015*). In addition, the beneficial root fungal endophyte *Serendipita indica* secretes ecto-5'-nucleotidases (E5'NT) that, like plant 5'NTs, are capable of hydrolyzing eATP and thereby shifting the equilibrium in the apoplast from eATP to eAdo (*Nizam et al., 2019*). Together, these data suggest that the eATP/eAdo balance is relevant for fungal infection and that microbes might manipulate the apoplastic Ado levels in its favor, as they do with the apoplastic pH ($pH_{apo}$) (*Felle, 2001*; *Masachis et al., 2016*; *Kesten et al., 2019*). However, the roles of eAdo and of the eATP/eAdo equilibrium in plant defense remain poorly understood.

*Fusarium oxysporum* (Fo) is one of the plant pathogenic fungi whose capacity to induce $pH_{apo}$ changes is best studied (*Masachis et al., 2016*; *Kesten et al., 2019*). As a microbe that mainly grows and advances through the apoplast, it represents an excellent model system to study plant-microbe molecular communication in this plant region. We thus made use of an elicitor mix, a crude mycelia extract of a lyophilized *Arabidopsis* pathogen Fo5176 suspension. This extract led to rapid and local alterations of the $pH_{apo}$ and cellulose synthesis machinery similar to those observed by the fungal hyphae (*Kesten et al., 2019*). Through fractionation followed by HPLC purification of the elicitor mix, we identified Ado as an abundant and active molecule in the Fo5176 elicitor mix. Our data indicate that Fo5176 increases the levels of eAdo in the apoplast to facilitate its growth in the host. Genetic, transcriptomic, and live-cell high-resolution microscopy approaches revealed that Ado alters ATP-induced plant defense responses.

## Results

### Fo5176 secretes Ado that seems to counteract eATP-induced plant defense

We recently showed that a Fo5176 elicitor mix regulates the growth-defense balance in plants (*Kesten et al., 2019*). To identify the molecules in the elicitor mix involved in this response, we performed a bioassay-guided fractionation, using a C18 solid phase cartridge with a 10% step gradient of a

**Table 1.** NMR data of Ado.
The chemical shifts of 1H and 13C found in Ado. Blanks are heteroatoms in the main chain. Selected COSY and HMBC correlations are included to demonstrate linkage.

| Atom | δ13C | δ1H | HMBC | COSY |
|---|---|---|---|---|
| 1 | | | | |
| 2 | 152.4 | 1H 8.13 (s) | 4, 6 | |
| 3 | | | | |
| 4 | 149.1 | - | | |
| 5 | 119.1 | - | | |
| 6 | 156.1 | - | | |
| 7 | | | | |
| 8 | 139.7 | 1H 8.34 (s) | 4, 5 | |
| 9 | | | | |
| 1' | 90.8 | 1H 5.95 (d, J=3.9Hz) | 4, 8, 2' | 2' |
| 2' | 81.1 | 1H 4.52 (dd, J=6.3, 3.9 Hz) | | 1', 3' |
| 3' | 77.1 | 1H 4.22 (dd J=6.3, 3.8 Hz) | | 2', 4' |
| 4' | 87.7 | 1H 3.93 (q J=3.8 Hz) | | 3', 5' |
| 5' | 62.4 | 1H 3.61 (m), 1H 3.50 (m) | | 5', 4' |

water:methanol solvent system followed by HPLC on a semiprep C18 column. This approach yielded a purified active component that we identified as adenosine (Ado) by standard 1D and 2D NMR (*Table 1*) and high-resolution mass spectrometry. Comparison of its retention time and mass data to a pure commercial standard, confirmed that Ado is a main component of elicitor mixes generated from in vitro-grown Fo5176 (*Figure 1A* ). We then asked whether the fungus secretes this potential new elicitor during plant infection. As the plant or fungal origin of the extracellular Ado (eAdo) present in the host apoplast cannot be distinguished, we tested the expression of genes required for hydrolysis and secretion of eAdo, *ENT,* and *5'NT*, respectively, in the host and the intruder during their interaction. Both *FoENT* and *Fo5'NT* were significantly upregulated during root colonization (*Figure 1B and C*), while the expression of the *Arabidopsis*' homologs was not altered by the presence of the fungus (*Figure 1—figure supplement 1*). These data are supported by the identification of Fo5'NT protein (g8638) in the secretome of Fo5176-infected roots of hydroponically grown Col-0 plants (*Gámez-Arjona et al., 2022*), indicating that Fo5176 might indeed secrete Ado to the apoplast while colonizing plant roots.

Considering the biochemical relation between Ado and ATP and the reported role of eATP in plant immunity (*Chen et al., 2017*; *Cao et al., 2014*; *Kumar et al., 2020*), we tested the putative influence of eAdo on ATP-induced plant defense. Thus, we first investigated if the plant response to Fo5176 is eATP-dependent by exposing the plants to different concentrations of ATP (10 µM to 500 µM) while infected by Fo5176, as described previously (*Kesten et al., 2019*; *Huerta et al., 2020*). Indeed, 300–500 µM ATP significantly reduced Fo5176 vascular colonization (*Figure 1—figure supplement 2*). To assess the effect of eAdo on ATP-induced plant defense, we exposed the roots to 500 µM ATP and Ado in equimolar to doubled concentrations of ATP (*Figure 1—figure supplement 2*). Plants treated with 1 mM Ado and 500 µM eATP were indistinguishable from control plants regarding vascular penetrations by Fo5176 (*Figure 1D* and S2B). Ado on its own did not have any detectable effect on fungal vascular penetration under our experimental conditions, indicating that Ado plays an important role in the plant eATP signaling regulation (*Figure 1D*). Importantly, eATP-induced root and fungal growth inhibition was not recovered by the addition of Ado (*Figure 1E–G*). These results indicate that the plant response to Ado is ATP-dependent and implicate a mechanism in which eAdo interferes with eATP-induced plant defense responses that is not based on plant- or fungal-growth retardation.

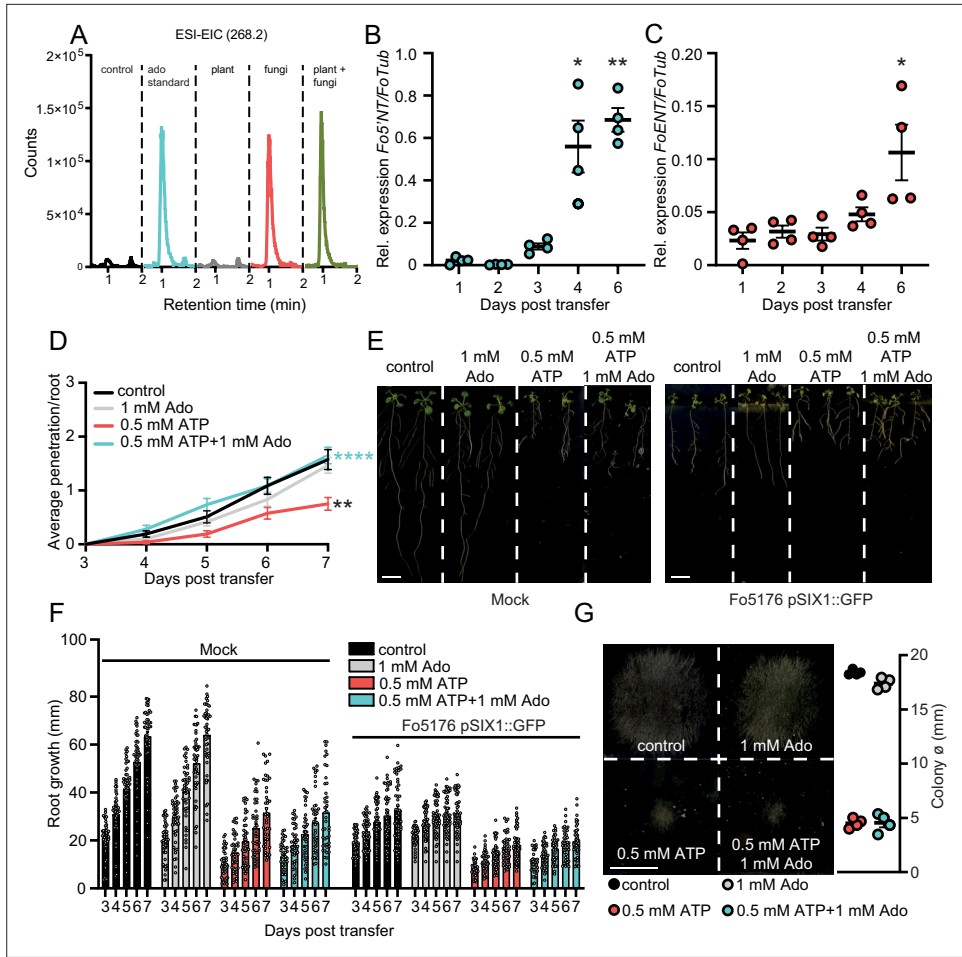

**Figure 1.** Enhanced apoplastic Ado counteracts eATP-induced reduction of fungal penetration on root vasculature. (**A**) Overlayed LC-MS extracted ion chromatograms of blank (red), adenosine (green), plant and fungus (blue).Overlayed LC-MS extracted ion chromatogram of blank (black), 250 ng adenosine (blue), fungi (red), plant (gray), plant and fungi (green) (**B**) and (**C**) *Fo5'NT* (**B**) and *FoENT* (**C**) expression relative to *FoTub* in hydroponically-grown *Arabidopsis* roots at various days post treatment (dpt) with Fo spores. Values are mean ± SEM, N≥20, Welch's unpaired t-test: (**B**) 1 dpt vs. 4 dpt: * p-value ≤0.05, 1 dpt vs. 6 dpt: ** p-value ≤0.01; (**C**) 1 dpt vs. 6 dpt: * p-value ≤0.05. (**D**) Cumulative Fo5176 pSIX1::GFP root vascular penetrations in wild-type (Col-0) seedlings at different days post-transfer to plates containing ½ MS (control) and 1 mM Ado and/or 0.5 mM ATP. Values are mean ± SEM, N≥52 from three independent experiments. RM two-way ANOVA with Tukey post-hoc test on control vs. 0.5 mM ATP: p≤0.001 (treatment), p≤0.001 (time), p≤0.0001 (treatment x time). Significant differences compared to control (black asterisk) and 500 µM ATP (blue asterisks) at 7 dpt are indicated on the graph (Tukey test); statistics of remaining time points are summarized in ***Supplementary file 2***. (**E**) Representative images of Col-0 seedlings at 7 dpt to mock (left) or Fo5176 pSIX1::GFP (right) plates. Scale bar = 1 cm. (**F**) Root growth of plants as shown in (**E**) at different days post transfer to mock or Fo5176 pSIX1::GFP-containing plates. Values are mean ± SEM, N≥52 roots from three independent experiments, RM two-way ANOVA p (treatment, time, treatment x time): control vs 500 µM ATP (≤0.0001, ≤0.0001, ≤0.0001); control vs 500 µM ATP +1 mM Ado (≤0.0001, ≤0.0001, ≤0.0001); control infected vs 500 µM ATP infected (≤0.0001, ≤0.0001, ≤0.001); control infected vs 500 µM ATP +1 mM Ado-infected (≤0.0001, ≤0.0001, ≤0.0001). (**G**) Colony diameters of Fo5176 grown for 4 days on plates containing ½ MS (control) and 1 mM Ado and/or 0.5 mM ATP. Values are mean ± SEM, N=4, Welch's unpaired t-test control vs 0.5 mM ATP: **** p-value ≤0.0001; control vs. 0.5 mM ATP +1 mM Ado: **** p-value ≤0.0001. Scale bar = 1 cm.

The online version of this article includes the following source data and figure supplement(s) for figure 1:

**Source data 1.** Enhanced apoplastic Ado counteracts eATP-induced reduction of fungal penetration on root vasculature.

**Figure supplement 1.** Plant *ENT3* and *NSH3* expression do not change in response to Fo5176.

**Figure supplement 1—source data 1.** Plant *ENT3* and *NSH3* expression do not change in response to Fo5176.

*Figure 1 continued on next page*

*Figure 1 continued*

**Figure supplement 2.** ATP-induced plant defense is counteracted by doubled concentration of eAdo.

**Figure supplement 2—source data 1.** ATP-induced plant defense is counteracted by doubled concentration of eAdo.

## Plants impaired in ATP sensing or with high eAdo/eATP levels are more susceptible to Fo5176

To further test the role of Fo-secreted Ado (*Figure 1A–C*) interfering with eATP during plant-pathogen interaction, we aimed at creating fungal mutants lacking *FoE5'NT* and *FoENT*. Although more than 165 potential transformants showed successful insertion of the resistance cassette into the fungal genome, none of them were knock-out mutants of the target genes, that is the cassette was inserted off-target. This indicates the importance of these genes for fungal viability and the very possible lethality of *FoΔE5'NT* and *FoΔENT* mutants. As manipulating the fungal eAdo levels was not successful, we addressed the influence of eAdo/eATP on plant-pathogen interactions from the plant's side using an *Arabidopsis* mutant altered in this ratio (*ent3nsh3*) (*Daumann et al., 2015*) or eATP

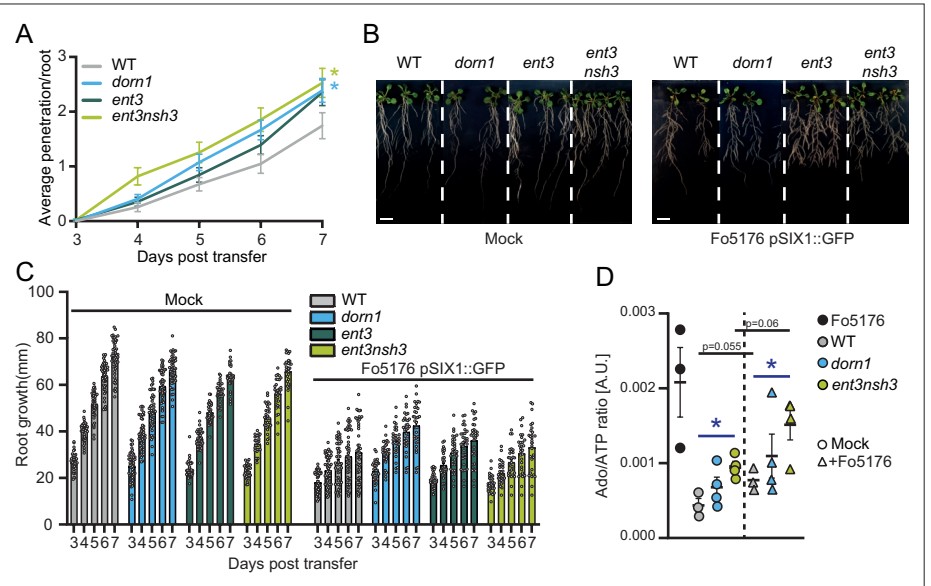

**Figure 2.** Increased extracellular Adenosine levels enhance fungal penetration rates. (**A**) Cumulative Fo5176 pSIX1::GFP root vascular penetrations in wild-type (WT; Col-0), *dorn1*, *ent3*, and *ent3nsh3* seedlings at different days post-transfer (dpt) to spore-containing plates. Values are mean ± SEM, N≥94 from three independent experiments. RM two-way ANOVA *P* (treatment, time, treatment x time) on WT vs. *dorn1* (≤0.05,≤0.0001,≤0.05); WT vs. *ent3nsh3* (≤0.0075,≤0.0001,≤0.0061). Significant differences compared to WT plants at 7 dpt are indicated on the graph (Tukey test); statistics of remaining time points are summarized in *Supplementary file 2*. (**B**) Representative images of 8-day-old mock or Fo5176 pSIX1::GFP infected plants as indicated in (**A**) at 7 days post-transfer to plates containing Fo5176 pSIX1::GFP spores. Scale bar = 1 cm. (**C**) Root growth of plants indicated in (**B**) at different days post transfer to mock or Fo5176 pSIX1::GFP-containing plates. Values are mean ± SEM, N≥79 from three independent experiments, RM two-way ANOVA *P* (genotype, time, genotype x time): WT vs. *dorn1* (≤0.01,≤0.0001,≤0.001); WT vs. *ent3* (≤0.0001,≤0.0001,≤0.0001); WT vs. *ent3nsh3* (≤0.0001,≤0.0001,≤0.05); WT infected vs. *dorn1* infected (≤0.0001,≤0.0001,≤0.0001); WT infected vs. *ent3* infected (≤0.05,≤0.0001,≤0.05). (**D**) Ado/ATP ratio content in media from 10 days-old hydroponically-grown wild-type (WT; Col-0), *dorn1*, and *ent3nsh3* seedlings at 4 days after transfer to media with (+Fo5176) and without (Mock) Fo5176 spores, and in media where Fo5176 was growing alone for 4 days (Fo5176). Values are mean ± SEM, N≥3 biological replicates, Welch's unpaired t-test in respect to their mock (black) or among genotypes (blue): * p-*value* ≤0.05.

The online version of this article includes the following source data and figure supplement(s) for figure 2:

**Source data 1.** Increased extracellular Adenosine levels enhance fungal penetration rates.

**Figure supplement 1.** Ado and ATP levels in the media are altered by Fo5176 infection.

**Figure supplement 1—source data 1.** Ado and ATP levels in the media are altered by Fo5176 infection.

sensing (*dorn1*) (*Choi et al., 2014a*). Double mutant *ent3nsh3* plants showed an increased suscepti-bility to fungal colonization, while the single mutant *ent3* was not significantly affected in its response to the pathogen (*Figure 2A*). Lack of DORN1 caused an increased fungal vascular penetration rate (*Figure 2A*), underlining the role of eATP as a DAMP in Arabidopsis-Fo5176 interaction and confirming previous data (*Kumar et al., 2020*). Compared to Fo5176-treated WT, *dorn1* and *ent3* plants showed significantly increased primary root growth over time, while *ent3nsh3* did not differ substantially from WT, despite its higher infection numbers (*Figure 2B and C*). These data indicate that the anticipated elevated apoplastic Ado/ATP ratio in *enth3nsh3* (*Daumann et al., 2015*) might have a main role in enhancing plant colonization by Fo5176. To test this hypothesis, we measured both soluble Ado and ATP levels in the media of hydroponically grown plants in control and Fo5176-infected conditions, as proxy for the levels of those molecules in the apoplast. As expected, the growth media of infected *ent3nsh3* plants showed significantly elevated Ado levels in comparison to mock treatments, which were not observed in WT or *dorn1* plants (*Figure 2—figure supplement 1*). In addition, we observed

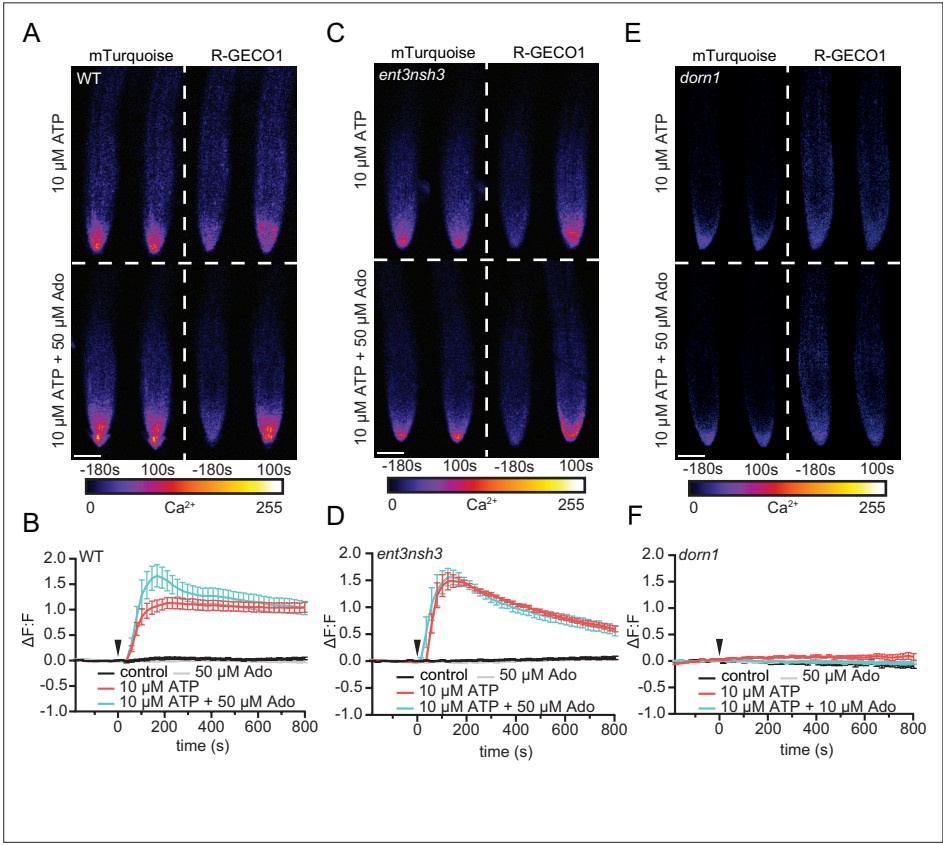

**Figure 3.** eAdo enhances eATP-induced DORN1-mediated cytosolic Ca$^{2+}$ peak. (**A**), (**C**), and (**E**) Representative images of five-days-old wild-type (WT; Col-0; **A**), *ent3nsh3* (**C**) and *dorn1* (**E**) roots expressing the cytoCa$^{2+}$ sensor, R-GECO1-mTurquoise −180 s before and 100 s after being exposed to ATP (upper panels) or ATP +Ado (bottom panels). Heatmaps indicate signal intensity (arbitrary units). Scale bar = 125 µm. (**B**), (**D**), and (**F**) cytoCa$^{2+}$ in roots as in (**A**), (**C**), and (**E**) represented as normalized fluorescence intensity changes (ΔF:F) of R-GECO1: mTurquoise. Imaging started 180 s before either ATP or ATP +Ado was added (0 min; arrow head). Values are means ± SEM, N≥18 from three independent experiments. RM two-way ANOVA *P* (treatment, time, treatment x time): (**B**) control vs. 10 µM ATP (≤0.0001,≤0.01,≤0.0001); control vs. 10 µM ATP +50 µM Ado (≤0.0001,≤0.01,≤0.0001); 10 µM ATP vs. 10 µM ATP +50 µM Ado (≤0.0001,≤0.0001,≤0.0001); (**D**) control vs. 10 µM ATP (≤0.0001,≤0.0001,≤0.0001); control vs. 10 µM ATP +50 µM Ado (≤0.0001,≤0.01,≤0.0001).

The online version of this article includes the following source data and figure supplement(s) for figure 3:

**Source data 1.** eAdo enhances eATP-induced DORN1-mediated cytosolic Ca$^{2+}$ peak.

**Figure supplement 1.** Extracellular adenosine increases extracellular ATP induced DORN1 mediated Ca$^{2+}$ influx.

**Figure supplement 1—source data 1.** Extracellular adenosine increases extracellular ATP induced DORN1 mediated Ca$^{2+}$ influx.

significantly lower amounts of ATP in *ent3nsh3* mock-media compared to all other tested genotypes (*Figure 2—figure supplement 1*), indicating that this mutant has a higher eAdo/eATP ratio than WT in control conditions that is preserved upon Fo5176 infection due to the increase on eAdo (*Figure 2D*).

## eAdo increases the rapid ATP-induced transient cytosolic Ca²⁺ peak

To molecularly characterize the high susceptibility of *dorn1* and *ent3nsh3* to Fo5176, we explored earlier cellular immune responses, starting with the eATP-induced cytosolic Ca²⁺ (cytoCa²⁺) peak (*Choi et al., 2014a*). Employing the ratiometric cytoCa²⁺ sensor R-GECO1-mTurquoise (*Waadt et al., 2017*), we first determined the minimal ATP concentration that led to a consistent increase of intracellular Ca²⁺ in the meristematic and elongation zone of *Arabidopsis* WT roots. 10 μM ATP were enough to consistently induce a cytoCa²⁺ peak (*Figure 3A and B*), as previously reported (*Demidchik et al., 2003*). After introgressing R-GECO1-mTurquoise into both mutant lines, we found that addition of ATP to *ent3nsh3* led to a Ca²⁺ spike in the first 3 min, 1.5 times higher than that detected in WT, which decays to WT levels, indicating that an increased eAdo/eATP proportion might modulate rapid eATP-induced responses (*Figure 3A–D*). Consistent with the role of DORN1 as the main eATP receptor, we detected no cytoCa²⁺ peak in *dorn1* upon ATP treatment (*Figure 3E and F*). Next, we investigated if the external addition of Ado can interfere with this signaling process by testing various Ado concentrations (*Figure 3—figure supplement 1*). While Ado did not induce any changes in the cytoCa²⁺ levels up to a concentration of 200 μM, we detected that Ado enhanced the eATP-induced cytoCa²⁺ spike transiently when the ATP:Ado ratio was at least 1:5 (*Figure 3B*). The cytoCa²⁺ spike did not further increase in response to higher eATP:eAdo ratios (*Figure 3—figure supplement 1*). Accordingly, the high eATP-induced cytoCa²⁺ peak observed in *ent3nsh3* did not further increase by

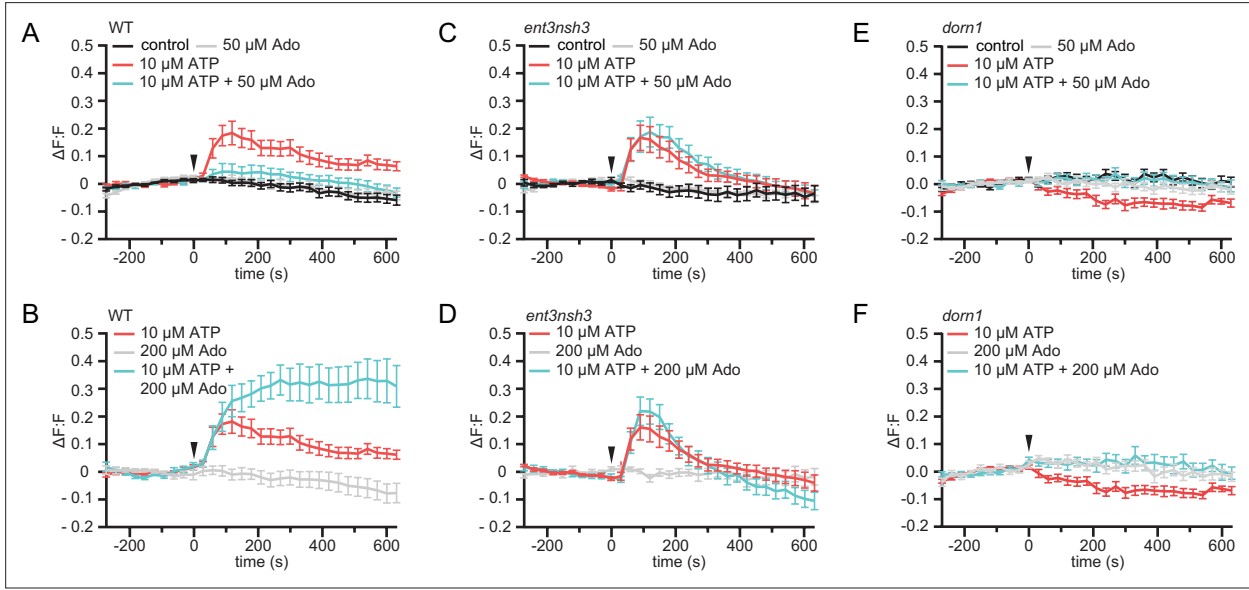

**Figure 4.** eAdo alters eATP-induced DORN1-mediated apoplast alkalization. (**A**), (**B**), (**C**), (**D**), (**E**), and (**F**) Apoplastic pH over time in roots expressing the pH_apo sensor SYP122-pHusion represented as the relative signal compared to the averaged baseline recorded prior to treatments (ΔF:F). Imaging started 270 s before either ATP or ATP +Ado was added (0 s; arrow head). Values are mean ± SEM; N≥12 seedlings from three independent experiments. RM two-way ANOVA, *P* (treatment, time, treatment x time) on (**A**) control vs. 10 μM ATP (≤0.0001,≤0.0001,≤0.0001); 10 μM ATP vs. 10 μM ATP +50 μM Ado (≥0.05,≤0.0001,≤0.05); (**B**) 200 μM Ado vs. 10 μM ATP +200 μM Ado (≤0.05,≤0.001,≤0.0001); 10 μM ATP vs. 10 μM ATP +200 μM Ado (≤0.01,≤0.01,≤0.0001); 200 μM Ado vs. 10 μM ATP (≤0.05,≤0.001,≤0.0001); (**C**) control vs. 10 μM ATP (≤0.05,≤0.0001,≤0.0001); control vs. 10 μM ATP +50 μM Ado (≤0.05,≤0.0001,≤0.0001); ATP vs. 10 μM ATP +50 μM Ado (≤0.01,≤0.0001,≤0.0001); (**D**) 200 μM Ado vs. 10 μM ATP (≥0.05,≤0.0001,≤0.0001); 200 μM Ado vs. 10 μM ATP +200 μM Ado (≥0.05,≤0.0001,≤0.0001); 10 μM ATP vs. 200 μM Ado +10 μM ATP; (**E**) control vs. 10 μM ATP (≤0.001,≤0.0001,≤0.0001); (**F**) 200 μM Ado vs. 10 μM ATP (≥0.05,≤0.001,≤0.01).

The online version of this article includes the following source data and figure supplement(s) for figure 4:

**Source data 1.** eAdo alters eATP-induced DORN1-mediated apoplast alkalization.

**Figure supplement 1.** Extracellular adenosine accumulation and absent extracellular ATP receptor DORN1 elevate apoplastic pH.

**Figure supplement 1—source data 1.** Extracellular adenosine accumulation and absent extracellular ATP receptor DORN1 elevate apoplastic pH.

adding Ado (*Figure 3C and D*; *Figure 3—figure supplement 1*). These data indicate that chemical or genetic enhancement of eAdo/eATP rapidly increases the eATP-induced transient cytoCa$^{2+}$ peak up to a certain eATP:eAdo concentration ratio. Moreover, we observed that eAdo could not alter the lack of response of *dorn1* to ATP (*Figure 3E and F*; *Figure 3—figure supplement 1*), confirming that the plant response to Ado is ATP-dependent.

## eAdo alters the ATP-induced apoplast alkalization

Exogenous application of ATP induces apoplast alkalization, as part of the fast plant response to DAMPs (*Wu et al., 2008*; *Hao et al., 2012*). Hence, we investigated if, as we observed for the cytoCa$^{2+}$ peak, eAdo also influences the ATP-dependent apoplastic pH (pH$_{apo}$) changes. By imaging the ratiometric pH$_{apo}$ sensor SYP122-pHusion (*Kesten et al., 2019*) in WT roots, we confirmed that the apoplast alkalizes in response to the same ATP concentration required to induce a cytoCa$^{2+}$ peak (10 µM; *Figure 4—figure supplement 1*), which we used concurrently for all further experiments. Analogous to the effect on cytoCa$^{2+}$ levels, Ado did not affect the pH$_{apo}$ on its own even at concentrations of 200 µM, while it altered the plant response when combined with eATP starting at 1:5 eATP:eAdo ratio (*Figure 4* and S5). eAdo concentrations up to 50 µM counteracted the eATP-induced apoplast alkalization, while 200 µM eAdo enhanced the eATP-dependent pH$_{apo}$ peak (*Figure 4A and B*, *Figure 4—figure supplement 1A*). On the other hand, *ent3nsh3* mutants showed comparable pH$_{apo}$ response to ATP as observed in WT roots, a response that was not altered by the Ado treatment (*Figure 4C and D*, *Figure 4—figure supplement 1B*). These results indicate that the high eAdo/eATP ratio in *ent3nsh3* apoplast cannot alter the plant response to eATP regarding pH$_{apo}$ changes but block the effect of exogenous Ado. In this context, it has to be highlighted that *ent3nsh3* mutants already show an elevated apoplastic pH under physiological conditions (pH = 6.00) in comparison to WT (pH = 5.54) (*Figure 4—figure supplement 1D*). Unexpectedly, *dorn1* responded to eATP with a slight, but significant, pH$_{apo}$ decrease that was restored to control levels by eAdo (*Figure 4E and F*; *Figure 4—figure supplement 1C*). Our data indicate that ATP induces a DORN1-independent apoplastic acidification, which seems to be counteracted by eAdo. Moreover, *dorn1* roots also showed a more alkaline apoplast than WT under control conditions, as detected in *ent3nsh3* (*Figure 4—figure supplement 1D*), hinting at a disturbed proton homeostasis in both mutants.

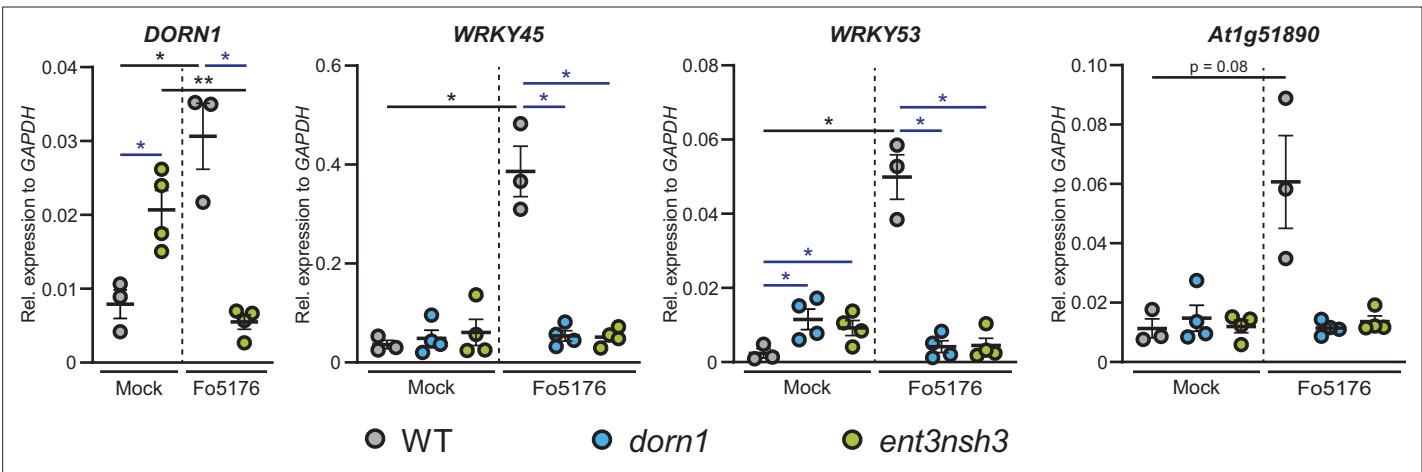

**Figure 5.** Accumulation of extracellular Ado impedes DORN1-mediated gene defense upregulation *DONR1*, *WRKY45*, *WRKY53*, and *At1g51890* expression relative to *AtGAPDH* in WT (Col-0), *dorn1*, or *ent3nsh3* roots 4 days post-treatment with Fo5176 spores or with control media (Mock). Values are mean ± SEM, N≥3 biological replicates, Welch's unpaired t-test within each genotype in respect to their mock (black) or among genotypes (blue); * p-value ≤0.05, ** p-value ≤0.01.

The online version of this article includes the following source data for figure 5:

**Source data 1.** Accumulation of extracellular Ado impedes DORN1-mediated gene defense upregulation.

## The expression of *Arabidopsis* defense genes in response to Fo5176 is eATP/eAdo-dependent

To further investigate the influence of Fo5176 on the activation of eATP/eAdo-dependent plant immune responses, we measured the expression of four defense-related genes upon Fo5176 infection. In agreement with the function of eATP as DAMP, *DORN1* expression increased in WT infected-roots but was significantly downregulated in *ent3nsh3* mutant plants upon Fo5176 colonization (*Figure 5*). The expression of three genes previously reported to be activated in Fo5176-infected *Arabidopsis* roots; *WRKY45*, *WRKY53,* and *At1g51890 Masachis et al., 2016*; *Kesten et al., 2019*; *Gámez-Arjona et al., 2022* followed a similar pattern as they were all upregulated in response to Fo5176 in WT plants but not in *dorn1* or *ent3nsh3* mutants (*Figure 5*). These data confirm that Fo5176 induces a eATP/eAdo-dependent plant immune response that might explain the high susceptibility of *dorn1* and *ent3nsh3* mutants to the fungus.

## Discussion

Ado is known as a key extracellular mediator of the animal immune response and its molecular activity in relation with eATP is increasingly recognized (*Antonioli et al., 2019*; *Silva-Vilches et al., 2018*). However, the knowledge of its role in plant-microbe interaction is very scarce. In this work, we show that an increased apoplastic Ado/ATP ratio enhanced plant susceptibility to the soil-borne pathogen Fo5176 and that the fungus benefits from this effect by modifying its metabolism in planta to raise the Ado concentration in the apoplast.

We identify Ado as a main elicitor of Fo when grown in vitro (*Figure 1A*). The transcriptional upregulation of the fungal but not of the plant eAdo producing molecular machinery during infection and fungal secretion of *Fo5'NT* during root colonization indicates that Fo exudes this molecule when colonizing roots (*Figure 1B, C*, *Table 1*, *Figure 6*). Therefore, our data expand the current knowledge on apoplastic effectors secreted by plant fungal pathogens (*Masachis et al., 2016*; *Wei et al., 2022*; *Xia et al., 2020*). The presence of Ado in the media during Fo colonization of *Arabidopsis* roots did not alter the host-microbe interaction on its own, but blocked the ATP-induced plant resistance, while not affecting fungal growth (*Figure 1D–G*, *Figure 1—figure supplement 1*). Hence, we deduce that eAdo interferes with the ATP-induced plant immune system activation upon fungal infection. Genetically encoded plant sensors for cytosolic $Ca^{2+}$ levels and apoplastic pH confirmed that 10 µM ATP induces pattern-triggered immunity in *Arabidopsis* roots, as we could detect an immediate $cytoCa^{2+}$ peak and apoplastic alkalization in response to this molecule (*Figures 3A–B , and 4A–B*). Importantly, the application of 50 µM Ado further enhanced the transient but not the sustained $Ca^{2+}$ influx response elicited by ATP, while Ado alone did not induce any plant response different from the control treatment (*Figure 3A and B*). A comparable mechanism was already discovered in oviductal ciliated cells in which adenosine itself is inactive but increases ATP induced calcium influx through activation of protein kinase A (*Barrera et al., 2007*). The same Ado concentration efficiently blocked the ATP-induced alkalization of the root apoplast, while Ado levels above a certain threshold significantly increased the ATP effect on apoplastic pH without any effect on the sustained $Ca^{2+}$ influx (*Figure 4A, B*, *Figure 4—figure supplement 1*). As apoplastic alkalization has been reported to be required for Fo pathogenesis (*Masachis et al., 2016*; *Kesten et al., 2019*), our data indicate that eAdo hinders ATP-induced plant resistance above a certain eATP/eAdo ratio by boosting the ATP-induced apoplast alkalization (*Figure 6*). Our short-term response data suggest that the rapid apoplastic alkalization generated by eATP is not a main contributor to plant defense against Fo5176, as (*Roux and Steinebrunner, 2007*) eAdo boosts this plant response potentially leading to increased fungal virulence (*Figures 1D and 4B*), and (*Chen et al., 2017*) eATP also generates a $pH_{apo}$ peak in the Fo5172 susceptible mutant ent3nsh3, similar to WT plants. It was previously reported that eATP induces defense-related gene expression independently of its effect on media pH (*Jewell et al., 2022*). However, more research is needed to clarify the relationship between pH changes and the defense mechanism triggered by eATP, and MAMPs/DAMPs/elicitors in general, in the process of an infection. The elevated $pH_{apo}$ of *dorn1* under mock conditions might indicate a positive regulatory function of signal transducers downstream of DORN1 on $H^+$-ATPases (*Figure 3—figure supplement 1D*). The reason for an ATP-dependent decrease in $pH_{apo}$ in *dorn1* mutants remains to be elucidated, although another target of eATP might be involved in that response.

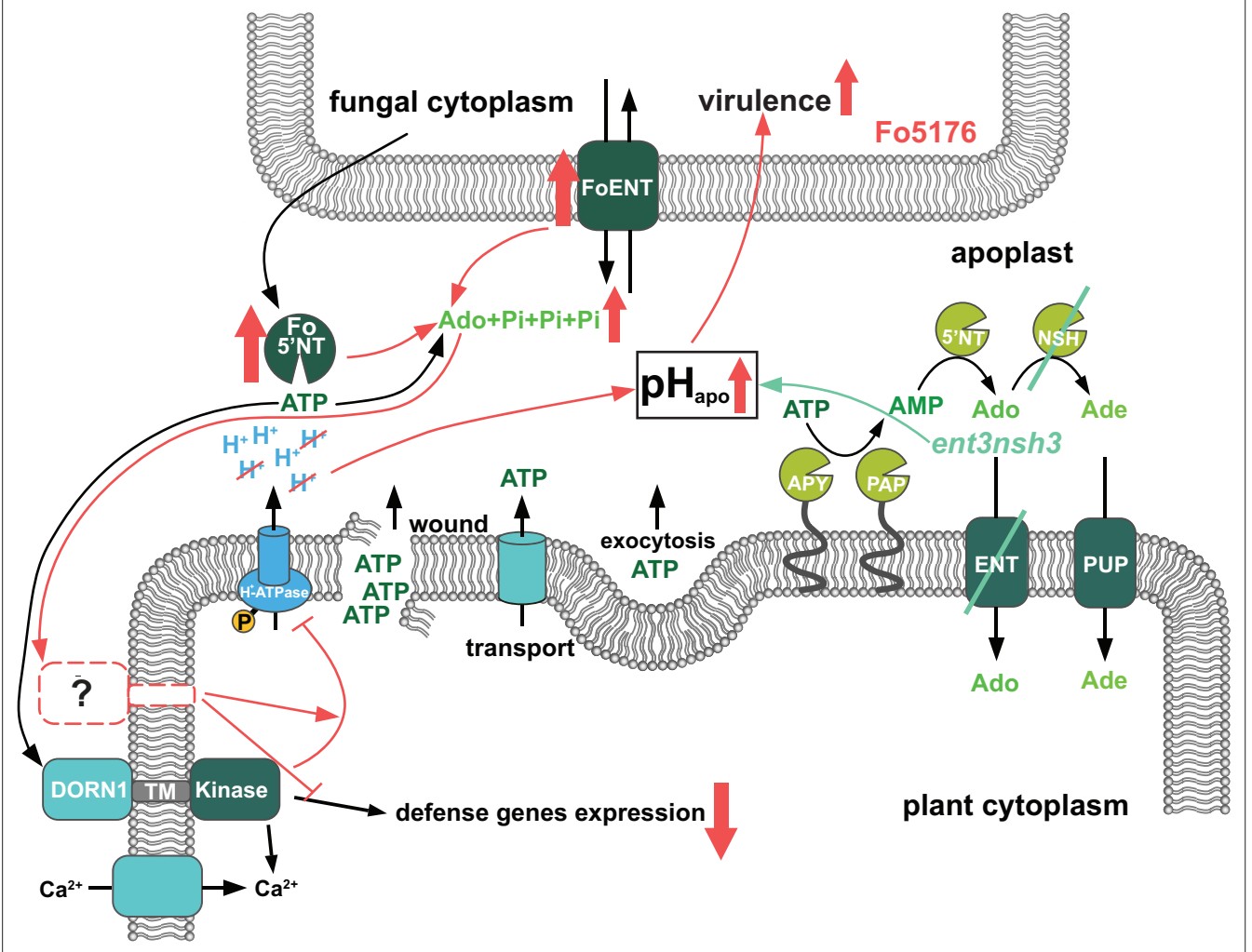

**Figure 6.** Scheme of the fungal-induced downregulation of the extracellular ATP and Adenosine homeostasis to increase its virulence. Previously published results (adapted from ***Nizam et al., 2019***) showed that ATP is released into the apoplast by wounding, active transport and exocytosis. Apyrases (APY) and purple acid phosphatases (PAP) degrade ATP to AMP, which is further processed by 5' nucleotidases (5'NT) and nucleoside hydrolases (NSH) to adenosine (Ado) and adenine (Ade). Equilibrative nucleotide transporters (ENT) and purine permease transporters (PUP) mediate take up of Ado and Ade into the cytoplasm. ATP is perceived by the puringeric receptor DORN1, which triggers $Ca^{2+}$ influx. Subsequently calcium dependent downstream signaling re-programming is initiated. First contact with *F. oxysporum* 5176 (Fo5176) elicits phosphorylation of *Arabidopsis* H+-ATPases (AHAs) and accordingly the apoplast acidifies (***Kesten et al., 2019***). In this work, we show that during root colonization (red symbols), Fo5176 upregulates expression of eATP hydrolyzing ecto-5' nucleotidase (Fo5'NT) and Ado transporter (FoENT) and secretes more Fo5'NT, increasing the extracellular Ado (eAdo) levels. eAdo interferes with ATP-induced plant immune system activation above a certain eAdo/eATP ratio by boosting ATP-induced apoplast alkalinization and thereby enhancing fungal virulence. Similar results were obtained in the *ent3nsh3* mutant where we detected higher eAdo/eATP levels compared to WT plants. Accordingly, this plant mutant is more susceptible to Fo5176 than WT plants and we observed a lack of upregulation of Fo5176-induced defense genes. Our data suggest that eAdo is perceived via a hitherto undiscovered dedicated plant receptor.

Infection assays revealed an increased susceptibility of *ent3nsh3* mutants compared to WT whereas *ent3* single mutants were not significantly different from WT (***Figure 2A***). We did not detect higher Ado levels in the media in contact with *ent3nsh3* roots compared to WT and *dorn1* lines (***Figure 2D***), confirming previous data showing similar eAdo levels in *ent3nsh3* and WT roots grown hydroponically (***Daumann et al., 2015***). Our results, though, indicate that the *ent3nsh3* mutant has a higher eAdo/eATP ratio in control media than WT plants, as it secretes comparable levels of Ado but less ATP to the media in mock conditions (***Figures 2D and 3***). This enhanced eAdo/eATP ratio might explain the constitutive upregulation of *DORN1* in *ent3nsh3* compared to WT (***Figure 5***). A potential higher activity of DORN1 in *ent3nsh3* under mock conditions can explain its increased response to ATP regarding cytosolic $Ca^{2+}$ peak and $pH_{apo}$ (***Figures 3C–D and 4C–D***). However, during Fo5176 infection,

those high *DORN1* expression levels detected in mock *ent3nsh3* drop, most probably as a result of the lack of its capacity to respond to Fo5176 infection, similar to that observed in the *dorn1* mutant (*Figure 5*). Thus, we conclude that a constitutively high eAdo/eATP ratio reduces plant resistance to Fo, an hypothesis substantiated by experiments in which WT plants showed similar responses when exposed to these molecules (*Figure 1D*). Furthermore, deficient defense responses on the transcriptional level in the *ent3nsh3* mutants corroborated this idea (*Figure 5*). The high cytoCa$^{2+}$ peaks detected in *ent3nsh3* in response to ATP, equivalent to what we observed in WT plants upon ATP +Ado, suggest a sufficient enrichment of eAdo/eATP in the mutant to respond to ATP (*Figure 3B and D*). These high ATP-induced transient cytoCa$^{2+}$ levels are not enough to activate a proper defense mechanism in *ent3nsh3*, in agreement with previously reported data (*Figure 5*, *Blume et al., 2000*). On the other hand, *ent3nsh3* pH$_{apo}$ changes in response to ATP were similar to those observed in WT, but the response was not changed by the addition of Ado, which altered the alkalization peak in control plants (*Figure 4A–D*). This could be explained by the conflicting published data about ENT3 being a proton symporter while transporting Ado (*Traub et al., 2007*; *Möhlmann et al., 2001*; *Wormit et al., 2004*; *Chen et al., 2006*). Since Ado requires the presence of ATP to alter the pH$_{apo}$ in WT plants, it is conceivable that the perception of ATP by DORN1 is required to initiate essential phosphorylation of ENT3 prior to transport, as reported for an ENT-family member in mammals (*Reyes et al., 2011*). In addition, since we still detected a pH$_{apo}$ increase in *ent3nsh3* mutants in response to ATP, we hypothesize that there is another proton-distribution-modifying component involved. Considering the alkaline apoplast detected in *ent3nsh3* under mock conditions (*Figure 4—figure supplement 1D*; *Haruta and Sussman, 2012*; *Behera et al., 2018*), we anticipate that a plasma-membrane localized H$^{+}$-ATPase might be negatively controlled by eAdo. This constitutive high pH$_{apo}$ measured in *ent3nsh3* might explain its different response to ATP +200 µM Ado compared to WT since the proton deficiency in the apoplast restricts the plant's ability to further increase pH$_{apo}$ (*Figure 4B and D*; S5A and B). It also has to be taken into account that the prevalence of DORN1 in *ent3nsh3* mutants is higher compared to WT (*Figure 5*), which could enable an enhanced induction of downstream signals like transient cytosolic Ca$^{2+}$ peak (*Figure 3D*). The constitutively high apoplastic eAdo/eATP ratio and pH$_{apo}$ detected in *ent3nsh3* concurs with an enhanced eATP-dependent pH$_{apo}$ increase and could explain the higher susceptibility of this mutant to Fo5176 (*Figure 6*). We anticipate similar functions in response to other pathogens, based on the reported positive role of DORN1/P2K1 in plant resistance to the soilborne fungal pathogen *Rhizoctonia solani* (*Kumar et al., 2020*).

Our data suggest that Ado could act as an antagonist and compete with ATP over the DORN1 receptor. However, this hypothesis was discarded by *Choi et al., 2014a* who showed no competitive inhibition of the DORN1 receptor by Ado. It can, however, not be fully excluded that Ado acts as a non-competitive or allosteric antagonist of ATP at the DORN1 receptor. It is also possible that eAdo may directly regulate apoplastic enzymes, e.g., ecto-apyrase/E-NTPDase, E5'NT, or another phosphatase, thereby indirectly controlling eATP homeostasis. A third option to explain the eAdo influence on ATP-mediated plant responses is the existence of an eAdo receptor whose activation interferes with the ATP-induced cascade (*Figure 6*). Importantly, maximum cytosolic Ca$^{2+}$ concentrations as well as pH$_{apo}$ peaks were detected 100 s after simultaneous application of ATP and Ado in all plant genotypes, including the *dorn1* mutant. Therefore we expect eAdo to prompt its effect at the plasma membrane level, like eATP (*Figure 3B and D*), and suggest the existence of a dedicated plant Ado receptor as described in animals. Indeed, G protein coupled receptors are reported to bind and sense adenosine in mammals and yeast (*Antonioli et al., 2019*; *Wang et al., 2021*). However, considering that AtDORN1 is not directly homologous to its mammalian counterpart (*Choi et al., 2014b*), an Ado-receptor analogous to the mammalian purinergic G protein coupled receptor class is unlikely. Further research is necessary to clarify the mechanism of eAdo perception and activity in plant response to ATP.

## Materials and methods
### Plant material and growth conditions

All *Arabidopsis thaliana* lines were in Col-0 background. The pH$_{apo}$ sensor line pub10::SYP122-pHusion, the calcium sensor line pub10::R-GECO1-mTurquoise, *dorn1-3*, *ent3-1* and *ent3nsh3* were published previously (*Choi et al., 2014a*; *Daumann et al., 2015*; *Kesten et al., 2019*; *Waadt et al., 2017*).

Seedlings throughout all experiments were grown upright on solid, non-buffered half MS media (pH 5.75) at 24 °C with a photoperiod of 16 hr for the indicated timeframes.

## Fungal material and growth conditions

*Fusarium oxysporum* Fo5176 and *Fusarium oxysporum* Fo5176 pSIX::GFP were used throughout this study. Strain culture and storage were performed as described earlier (*Di Pietro et al., 2001*). Fo5176 was grown in liquid half potato dextrose broth (PDB) at 27 °C for 5 days in the dark. Spores were collected by filtering the suspension through miracloth, centrifuging the filtrate at 3500 rcf, discarding the supernatant and resuspending the spores in $dsH_2O$.

## Fungal elicitor mix preparation, and fractionation, and molecule identification

Fungal elicitor mix was prepared as published previously (*Kesten et al., 2019*; *Baldrich et al., 2014*) and separated via enrichment using Discovery DSC-C18 (2 g) columns (Merck) with a $H_2O$/MeOH gradient from 100 % to 0 % $H_2O$ in 10% steps. Fractions were bioassayed and active fractions purified to individual components via an Agilent 1100 HPLC using Zorbax SB-C18 (9.4x150 mm) semi-prep column in a linear gradient of $H_2O$/MeOH and flow rate of 5 mL/min. Individual peaks were assayed for activity. The pure active compound was characterized by standard 1D and 2D NMR experiments performed at the NMR Service of the Laboratory of Organic Chemistry at ETH Zürich. All experiments were performed using d6-DMSO in a 600 MHz Bruker NMR equipped with a 5 mm probe. Data was analyzed using MestreNova 8.1 software (Mestrelab Research, Spain). LC-MS data was obtained on an Agilent 6400 LC-qTOF in scanning positive mode to produce a single signal with an m/z of 268.1044 (C10H13N5O4 calc. 268.1046 1.72 ppm) and identical retention time to an external standard of adenosine.

## Fungal transformation

PCR and complementary primers (*Supplementary file 1*) were used to generate two DNA fragments with overlapping ends (*Ho et al., 1989*). A resistance cassette containing the neomycin phosphotransferase (npTII) cloned between the *A. nidulans* gdpA promoter and the trpC terminator (*López-Berges et al., 2009*) was used to generate two DNA fragments promoting the homologous recombination in fungal protoplasts. Protoplasts were produced as described previously (*Powell and Kistler, 1990*) and their transformation done as reported by *Malardier et al., 1989*.

## In-vitro growth assay of Fo5176

Freshly harvested Fo5176 spores were diluted to $10^4$ spores/mL and 15 µL of it distributed on solid half MS plates containing 1 mM Ado, 0.5 mM ATP, or both, or none (control). After four days under the plant growth conditions described above ('Plant material and growth conditions'), the colony diameters were measured using FIJI (*Schindelin et al., 2012*).

## Plant plate infection assays

Plate infection assays were performed as described earlier (*Kesten et al., 2019*; *Huerta et al., 2020*). Ado and ATP treatment plates were generated by mixing hand-warm half MS media, 0.9% agar, with the specific amount of stock solution. Root growth was measured using FIJI (*Schindelin et al., 2012*).

## Hydroponic infection assay

Hydroponic infection assays were performed as previously described (*Menna et al., 2021*). Thirty seeds were grown on a foam floating on 50 mL liquid ½ MS media, pH 5.75, 1% sucrose. After 7 days the media was replaced by ½ MS without sucrose. Samples supposed to be infected were inoculated with $5*10^6$ Fo5176 spores. After the indicated days post transfer to spore-containing media, roots and fungal hyphae were harvested for subsequent expression analysis and the media was filtered. For ATP quantification media was flash frozen in liquid nitrogen, for Ado quantification it was freeze dried.

## Media ATP quantification

ATP levels in media from hydroponic infection experiments of hydroponically-grown plants were analyzed using the ATP Colorimetric/Fluorometric Assay Kit (Sigma, USA) and an Infinite M1000 plate

reader (Tecan, Switzerland). Assays were done as described in the manual and ATP was detected fluorescently. All samples and standards were measured in duplicates.

## Media Ado quantification

Freeze-dried media samples from hydroponic infection experiments were resuspended in 4 mL MilliQ water. The resulting mixture was loaded onto a 100 mg Discovery DSC-18 column (Supelco, USA). The column was eluted with 1 mL MilliQ water, 1 mL 70% MilliQ water with MeOH and finally 100% MeOH. The resulting aqueous elution was analyzed in positive mode using an Agilent 1200 Infinity II UPLC separation system coupled to an Agilent 6550 iFunnel qTOF mass spectrometer (Agilent, USA). Compounds were separated by infecting 5 µL of sample onto a Zorbax Eclipse Plus C8 RRHD UPLC column (2.1x100 mm, 1.8 µm) held at 50 °C and eluting with a linear water:acetonitrile (both modified with 0.1% formic acid) gradient, 99% water to 99% acetonitrile. Mass spectral data was acquired in positive mode with an electrospray ionization source and scanning a mass range of 100–2000 m/z. Quantification was done by integrating the m/z values corresponding to Ado in MassHunter Quantitative Analysis Software and compared to a standard curve generated at the time of sample measurements.

## In vitro growth assay of Fo5176

Freshly harvested spores were diluted to $10^4$ spores/mL and 15 µL of it distributed on solid half MS plates containing 1 mM Ado, 0.5 mM ATP or both. After four days under plant growth conditions the colony diameters were measured using FIJI.

## Gene expression analysis by real-time quantitative PCR

Freeze-dried fungal and plant material from plate infection assays respectively hydroponics was ground to powder using glass beads and a TissueLyser II (Quiagen, Netherlands). Total RNA was extracted using GENEzol reagent (Geneaid, Taiwan) following the manufacturer's protocol. One µg of RNA was used to generate first strand cDNA using the Maxima First Strand cDNA Synthesis-Kit (Thermo Scientific, USA) following the manufacturer's instructions. To amplify corresponding cDNA sequences primers (; *Choi et al., 2014a*; *Masachis et al., 2016*; *Kesten et al., 2019*; *Czechowski et al., 2005*; *Arnaud et al., 2017*; *Van der Does et al., 2017*) were used along with Fast SYBR Green Master Mix (Thermo Scientific, USA) under following cycle conditions: 95 °C for 3 min, 40 cycles of 94 °C for 10 s, 58 °C for 15 s and 72 °C for 10 s. Two technical replicates were performed for each reaction and the reference genes *AtGAPDH* and *Foβtub* were amplified on each plate for normalization. Relative expression was analyzed using the $2^{-\Delta Ct}$ method (*Schmittgen and Livak, 2008*).

## Ratiometric pH$_{apo}$ sensor imaging

Experiments were carried out as described earlier (*Kesten et al., 2019*). A Leica TCS SP8-AOBS (Leica Microsystems, Germany) confocal laser scanning microscope equipped with a Leica 10×0.3 NA HC PL Fluotar Ph1 objective or a Leica Stellaris 8 equipped with a Leica HC PL APO CS2 10 x/0.40 DRY were used. pHusion was excited and detected simultaneously (Excitation: GFP 488 nm, mRFP 561 nm; Detection: GFP between 500 and 545 nm; mRFP between 600 and 640 nm). Five-day-old *A. thaliana* seedlings expressing the sensor SYP122-pHusion grown on ½ MS +1% sucrose were transferred to imaging chambers as described previously (*Krebs and Schumacher, 2013*) but placed on top of 1% agarose cushions. Subsequently, the chamber was filled with ½ MS, pH 5.75. Images were collected as XYt series for 15 min with a time frame of 30 s. Image settings were kept identical throughout the experiments for each reporter line. After a recovery time of 15 min the experiment was started by acquiring ten images of the seedling' roots without treatment to create a baseline of averaged relative signal. Roots were imaged from the tip including their elongation zone. The different treatments were applied in a volume of 100 µL after 300 s. ΔF:F values were calculated according to following formula: $\Delta F : F = \frac{relative\ signal - baseline}{baseline}$ . Maximal amplitudes of ΔF:F signals were obtained by averaging the maximal ΔF:F signals of all curves. To collect standard curves for the pHapo ratiometric sensor, a set of nine buffers from pH 4.8 to pH 8.0 were used. Each buffer was based on 50 mM ammonium acetate. Buffer pH 4.8 comprised additionally 22 mM citric acid, 27 mM trisodium citrate. pH was adjusted with 0.01 M HCl. Buffers pH 5.2 to pH 6.4 contained 50 mM 2-(N-morpholino)ethanesulfonic acid (MES), buffers pH 6.8 to pH 8.0 were composed of 50 mM 4-(2-hydro- xyethyl)–1-piperazineethanesulfonic acid (HEPES). One M Bis-Tris propane was used to adjust the pH values of buffers pH 5.2 to pH 8.0.

Six to eight seedlings per buffer were incubated for 15 min and imaged after transfer to microscope slides.

## Ratiometric cytoCa$^{2+}$ sensor imaging

Imaging was done as described for the pH$_{apo}$ sensor (*Kesten et al., 2019*) with slight modifications. Five-day-old *A. thaliana* seedlings expressing the reporter R-GECO1-mTurquoise (*Waadt et al., 2017*) were grown on ½ MS, pH 5.75, 1% sucrose. mTurquoise was excited with 405 nm and detected between 460–520 nm, R-GECO1 was excited with 561 nm and detected between 580 and 640 nm. Imaging time frame was set to 20 s. Corrective flat field images for 405 nm were acquired by using 7-Diethylamino-4-methylcoumarin (Sigma D87759-5G, 50 mg/mL in DMSO). Relative signal was calculated by dividing mean gray values of the R-GECO1 channel by the mean gray values of the mTurquoise channel. ΔF:F values and Maximal amplitude of ΔF/F signals were calculated as described for the pH$_{apo}$ sensor.

## Statistical analyses

All statistical analyses were performed using Prism 9. Statistical methods and the resulting *P*-values are defined in the corresponding figure legends. Outlier tests were performed on datasets with. If the automatically detected fluorescent ratios of the genetic pH or Ca$^{2+}$sensors were measured to be outside of the standard curve range, they were excluded from the analysis. Such cases could always be allocated to severe drift of analyzed roots in the analysis chamber.

## Acknowledgements

Live cell imaging was performed with equipment maintained by the Scientific Center for Optical and Electron Microscopy (ScopeM, ETH Zurich) and by the Center for Advanced Bioimaging (CAB) Denmark. Funding: The work described in this manuscript was supported by the Peter und Traudl Engelhorn-Stiftung foundation, the ETH Foundation (SEED-05 19–2), the Novo Nordisk Foundation (Emerging Investigator grant NNF20OC0060564), and the Lundbeck foundation (R346-2020-1546) to CK; the Swiss National Science Foundation (grant 31003 A_182625) to CZ; a postdoctoral fellowship from the European Molecular Biology Organization (EMBO LTFs no. 683–2018) to JD.; the ETH Zurich core funding to CMDM, and the ETH Zurich core funding and the Swiss Swiss National foundation (SNF 310030_184769) to CSR.

## Additional information

### Funding

| Funder | Grant reference number | Author |
|---|---|---|
| Peter und Traudl Engelhorn Stiftung | | Christopher Kesten |
| ETH Zürich Foundation | SEED-05 19-2 | Christopher Kesten |
| Novo Nordisk Foundation | NNF20OC0060564 | Christopher Kesten |
| Lundbeck Foundation | R346-2020-1546 | Christopher Kesten |
| Swiss National Science Foundation | 31003A_182625 | Cyril Zipfel |
| European Molecular Biology Organization | 683-2018 | Julian Dindas |
| Swiss National Science Foundation | 310030_184769 | Clara Sanchez-Rodriguez |
| ETH Zürich | | Consuelo M De Moraes Clara Sanchez-Rodriguez |

The funders had no role in study design, data collection and interpretation, or the decision to submit the work for publication.

## Author contributions
Christopher Kesten, Conceptualization, Data curation, Formal analysis, Supervision, Investigation, Methodology, Writing – original draft, Writing – review and editing; Valentin Leitner, Data curation, Formal analysis, Investigation, Methodology, Writing – original draft, Writing – review and editing; Susanne Dora, James W Sims, Data curation, Formal analysis, Investigation, Methodology, Writing – review and editing; Julian Dindas, Data curation, Investigation; Cyril Zipfel, Supervision, Writing – review and editing; Consuelo M De Moraes, Supervision, Project administration; Clara Sanchez-Rodriguez, Conceptualization, Resources, Supervision, Methodology, Writing – original draft, Project administration, Writing – review and editing

## Author ORCIDs
Valentin Leitner ⓘ https://orcid.org/0000-0003-4984-5875
Susanne Dora ⓘ http://orcid.org/0000-0001-5411-9072
Cyril Zipfel ⓘ http://orcid.org/0000-0003-4935-8583
Consuelo M De Moraes ⓘ http://orcid.org/0000-0001-6737-9842
Clara Sanchez-Rodriguez ⓘ http://orcid.org/0000-0003-0987-9317

## Decision letter and Author response
Decision letter https://doi.org/10.7554/eLife.92913.sa1
Author response https://doi.org/10.7554/eLife.92913.sa2

---

# Additional files

## Supplementary files
• Supplementary file 1. Primers used in this study.

• Supplementary file 2. Statistical analysis of Fo5176 pSIX1::GFP root vascular penetrations in the indicated Arabidopsis genotypes.

## Data availability
All data generated or analysed during this study are included in the manuscript and supporting files.

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
