## [Editor Report]

This important paper reports how a fungal pathogen utilizes adenosine to perturb plant disease resistance and immune signaling. The authors convincingly show a key role of Ado/eATP in the alteration of apoplastic pH and pathogenesis. The research presented provides a foundation for the study of extracellular adenosine on purinergic signaling during plant-pathogen interactions.

---

## [Decision Letter]

**Decision letter after peer review:**

[Editors’ note: the authors submitted for reconsideration following the decision after peer review. What follows is the decision letter after the first round of review.]

Thank you for submitting the paper "Vascular fungi alter the apoplastic purinergic signaling in plants by deregulating the homeostasis of extracellular ATP and its metabolite adenosine" for consideration by *eLife*. Your article has been reviewed by 3 peer reviewers, one of whom is a member of our Board of Reviewing Editors, and the evaluation has been overseen a Senior Editor. The following individual involved in the review of your submission has agreed to reveal their identity: Kiwamu Tanaka (Reviewer #3).

Comments to the Authors:

We are sorry to say that, after consultation with the reviewers, we have decided that this work will not be considered further at this stage for publication by *eLife*. The requested experiments would likely exceed a normal time frame for a revision and it is our policy to prevent authors from multiple rounds of revisions with an uncertain outcome. We would, however, be willing to consider a revised manuscript that has fully addressed the main criticisms raised by reviewers.

The enhanced calcium signal elicited by by adenosine contradicts the increased susceptibility, and all reviewers believe this is a major unresolved issue. Furthermore, the assumption that Fusarium oxysporum produces adenosine to enhance penetration is not fully supported by the data, as pointed out by reviewer #3. In addition, the use of ent3 nsh3 double mutant is questioned because the mutant has the same amount of adenosine, which weakens the claim.

*Reviewer #1 (Recommendations for the authors):*

The authors identified Ado as an active elicitor produced by F. oxysporum and show that fungal genes, but not plant genes, controlling the production of Ado are specifically induced during infection. Like many elicitors, Ado likely plays a role in virulence, as the addition of Ado can block ATP-induced protection of plants against the fungal pathogen. Similarly, the authors show that the ent3 nsh3 double mutant which has an elevated eAdo/eATP ratio displayed increased susceptibility to the fungus. The authors provide data to show that eAdo can alter eATP signaling in a manner dependent on the ratio. Overall the work presents novel findings that homeostasis of eAdo and eATP plays an important role in plant resistance/susceptibility to the pathogen, which adds a twist to our understanding of the plant immune system. The limitation is with the underlying mechanism of disease susceptibility caused by eAdo/eATP change, as ca^2+^ signaling and apoplast alkalization investigated in this study do not appear to explain the phenotype of plants.

Some of the results observed can be confusing to readers and the authors need to clearly discuss. The alterations of eATP-triggered cytosolic ca^2+^ and apoplastic alkalization by eAdo do not explain the virulence function of eAdo. Ado apparently enhances the ATP-induced ca^2+^ at an eAdo/eATP ratio of 5:1. As ca^2+^ increase is known to be essential for plant immunity, the particular effect of Ado on ca^2+^ described here contradicts our knowledge of ca^2+^ signal in immunity. eAdo blocks eATP-induced alkalization, and the authors claim that blocking eATP-induced pH increases explains the virulence function of eAdo. Although apoplast alkalization is a well-known PTI response, whether and how this impacts disease resistance is not known. In addition, the ent3 nsh3 mutant has a high apoplast pH but is susceptible to F. oxysporum. Thus the authors need to be careful in interpreting the data. It is fine to use ca^2+^ and alkalization as a proxy to show that the eAdo/eATP ratio is important for the alteration of ATP signaling, but at the same time, the authors need to tell readers that these responses do not explain resistance/susceptibility! Thus the authors are strongly encouraged to tone down their statement and address the limitation of their findings.

*Reviewer #2 (Recommendations for the authors):*

This work on purinergic signaling in plants, and the influence of a pathogenic fungus on this represents a highly interesting topic. The combination of calcium response quantification with mutants involved in eATP signalling and control of apoplastic adenylate levels allows a deeper insight into this complex process.

It is shown that eAdo exerts different responses, it increases calcium responses, and leads to alkalization of the apoplast but counteracts the effect of eATP on root penetration. It is proposed that eAdo activates the plant's immune response, but root growth in the absence or presence of the pathogen is not affected by this metabolite. This is a complex scenario and I have difficulties following the authors' hypothesis. A scheme showing how eAdo interacts with eATP for receptor activation, and how apoplast pH and removal of eAdo is achieved could be helpful in explaining this better.

I miss the verification of the mutants that were reidentified (dorn1, ent3) and the crossed line ent3/nsh3. Were the same T-DNA insertion lines used as in the corresponding original publications? Please give this information in the methods section together with the line identifiers. Also please show a PCR check on cDNA.

Line 72ff: What does it mean, ado is the main component in elicitor mixes.

What is an elicitor mix? How many substances were detected in total? Is the method quantitative? Which percentage is ado then?

Difficult to see the colors in figure 1A, optimize the presentation.

Line 101: Why do the authors write "DORN1 impairment", isn't it a knockout?

Figure 2 D, E use of different symbols in addition to only colors would allow for easier access to the data.

*Reviewer #3 (Recommendations for the authors):*

The topic of the authors' study has a great impact and provides new insight into the purinergic signal and its homeostasis involved in plant-pathogen interactions.

Strengths:

Homeostasis of eATP has been studied for root growth, gravitropism, stomata opening/closure, and plant-symbiont interactions, while very limited information on this topic is currently available for plant-pathogen interactions. The research by Kesten et al. suggested an important role of the ATP level in apoplast that was targeted by the fungal pathogen Fusarium oxysporum. The authors proposed that the fungal pathogen releases Ado to regulate the eATP-induced plant defense during infection. There are only a few case studies regarding the Ado function in plants by Roux's group and Mohlmann's group where its mode of action has long been unclear as a negative regulator of the eATP signaling. Thus, the topic in the authors' manuscript has great potential to open up a new field of study in a regulatory mechanism of purinergic signaling in plants.

Weaknesses:

The discussion in the manuscript was incoherent and scarce, especially regarding their inconsistent results. For example, Ado inhibited the eATP-induced defense, whereas it further enhanced eATP-induced calcium elevation and alkalinization. An experiment was conducted under the assumption of high Ado in a mutant (ent3nsh3), while the actual level of Ado in the mutant was comparable with that in the wild-type plant. The authors need to make convincing explanations.

I would like to support the study in the manuscript by Kesten et al., given the impact of the topic and the scientific merits of the authors' discovery of the novel function of Ado in plant-pathogen interaction. The results appear to be preliminary and incomplete for publication at the present time. In addition, the discussion in the manuscript was incoherent and scarce.

The experiments using the ent3nsh3 mutant were performed under the assumption of the high level of endogenous Ado accumulated in the mutant. If this is the case, the effect of eATP could be nullified without the exogenous application of Ado. Moreover, Figure 2D showed that it was not actually a high level of Ado accumulated in the mutant. The authors are also recognizing that the eAdo level in ent3nsh3 was not remarkably different in comparison to WT (Line 229-231). Unfortunately, all experiments using the mutant are not actually supporting the authors' conclusions in Figure 2-5.

The authors did applaudable work isolating Ado as a potential inhibitor of eATP signaling out of a compound mix secreted from the fungal pathogen. However, validation of the Ado secretion from the fungi was not substantially demonstrated. Data only with gene expression measurement (Figures 1B and 1C) are just suggestive but not conclusive. Does the enzymatic activity of Fo5'NT (or FoENT) always correlate with its transcription? Does the protein level correlate with the transcription level of a fungal enzyme gene? Does an experiment using an isotope directly support the authors' conclusion?

It is well known that elevation of cytosolic calcium level is important to induce plant defense response. If eAdo is hypothesized to suppress the eATP-induced defense response, it is expected to reduce the calcium response. However, the authors showed that Ado further enhanced the calcium response, which is not consistent with Ado's inhibitory effect on eATP-induced defense response against the fungal pathogen. It does not make sense at all and the authors did not discuss it in the manuscript. Is it possible to demonstrate that the increased cytosolic calcium level suppresses the defense response and genuinely contributes to the susceptibility against fungal infection?

Since eAdo receptors have been reported and well studied in animals, it could be natural to adopt the idea of the eAdo function in plants as proposed by the authors in the Discussion section. However, there are still other possibilities to be tested on how eAdo suppresses eATP signaling. Ado could act as an antagonist and compete with ATP over the DORN1 receptor. It is also possible that eAdo may directly regulate apoplastic enzymes, e.g., ecto-apyrase/E-NTPDase, E5'NT, or another phosphatase, thereby indirectly controlling eATP homeostasis.

I recommend changing the title since "Vascular fungi" provides us the impression that the authors study the eATP signaling in the plant vascular tissue. "Ascomycete fungus" or "Soilborne fungus" would be appropriate for this study given their pathogenicity.

Line 58: The role of eATP against "different pathogens" was summarized in Table 1 of Jewell et al. https://doi.org/10.1093/plphys/kiac393, which could be an appropriate paper to cite.

Since this manuscript is about the role of eATP as a DAMP against the soilborne pathogen, it would be fair to compare the authors' data with the previous report where the soilborne pathogen Rhizoctonia (Kumar et al. https://doi.org/10.3389/fpls.2020.572920) was used and discuss further.

The term "elicitor" is about a compound that induces (elicits) plant defense. Toxin or effector is a compound that manipulates plant immune systems to help infection of symbionts and pathogens. The term should be distinguished carefully. In addition, I do not agree with defining "eAdo as a MAMP" on lines 203-4.

Since the study in this manuscript is about an apoplastic effector (toxin?) compound by fungal pathogens, the following 3 literature should be introduced together as previous precedents: Masachis et al. https://doi.org/10.1038/nmicrobiol.2016.43; Wei et al. https://doi.org/10.1038/s41467-022-29788-2; Xia et al. https://doi.org/10.1073/pnas.2012149117.

Since it is about [ATP] vs. [ATP + Ado], the statistic analysis should be performed between them for Fig1D and Figure 1-supl2.

Not everyone is an expert in the experiment of vascular penetration by fungal hyphae. It would be helpful for the broad audience if the authors kindly show a few representative pictures or diagrams to explain the method.

Why the authors used non-buffered media? ATP is a strong anion and it might have a significant impact on pH in the media. Root growth is known to be sensitive to pH. Maybe the Ado effect was masked by the effect of the pH lowering by the ATP addition. What would be the result if the experiment in Figure 2C is performed by preventing pH changing with a buffer to observe a true effect of ATP as a signal?

Line 137-8: I do not agree that the response to 10 μm ATP is enough to say "more sensitive than previous report". A previous report showed that even 3 μm eATP provided enough consistent data (Demidchik et al. https://doi.org/10.1104/pp.103.024091).

Line 161: It is inappropriate to mention "up to 100 uM" without such data.

Line 166: It is inappropriate to mention "high eAdo/eATP ration in ent3nsh3 apoplast without such data.

Line 176: How many genes did the authors test for "various" defense-related genes? Can it be specified?

Line 196: "just" 10 μm ATP? What are the authors emphasizing?

Line 199 and 200: My understanding is that the inhibition effect of Ado may be not due to a chemical reaction. So, "Ado itself was inactive" should be rephrased.

Line 205-7: The sentence does not make sense. How can eAdo block plant defense by blocking eATP-induced alkalinization even though alkalinization is for fungal pathogenesis? Instead, Figure 4B, where Ado further enhanced alkalinization, seems reasonably consistent with the previous case in ref 23.

Line 220: Since ATP is a strong anion, the acidification could be due to its chemical feature since no alkalinization in dorn1 was induced, as an alternative.

Line 221: The authors pointed out that P2K2 is involved in eATP-induced apoplastic acidification in Figures4E and 4F. Please explain and discuss further in detail.

Line 221-2: I had a tough time to understand how GPA is involved in the acidification in dorn1. Please elaborate more.

Line 247: I am not sure that it is a concentration-dependent manner since 10 μm and 200 μm of Ado evoked the opposite responses (Figure 4A and 4B). Please rephrase in an appropriate manner.

Line 249-51. I do not understand the sentence. Did the authors mean ENT3 is directly regulating (or transporting) protons since the authors speculated that phosphorylation of ENT3 by eATP is required for Ado-elicited pH changes? The following sentence on lines 251-3 was more confusing. Do the authors hypothesize ENT3 as an H^+^ transporter?

Line 252-55: The authors speculate that there are H^+^-ATPases for ATP/Ado-induced pH changes. Indeed, ATP-induced calcium response and apoplastic pH (also cytosolic pH) are reported to be associated based on the data in Behera et al. https://doi.org/10.1105/tpc.18.00655 and in Figure 5D in Haruta et al. https://doi.org/10.1104/pp.111.189167. It would be nice to discuss this further by citing these publications.

I do not understand the logic of the authors' conclusion on lines 262-3. The authors speculated that there is a potential Ado receptor because eATP signaling is required for the inhibition effect by Ado.

[Editors’ note: further revisions were suggested prior to acceptance, as described below.]

Thank you for submitting your article "Soil-borne fungi alter the apoplastic purinergic signaling in plants by deregulating the homeostasis of extracellular ATP and its metabolite adenosine" for consideration by *eLife*. Your article has been reviewed by 3 peer reviewers, and the evaluation has been overseen by a Reviewing Editor and Jürgen Kleine-Vehn as the Senior Editor.

Essential revisions:

1) Please correct/clarify the terminology pointed out by reviewer #2.

2) Cite references 41 and 42 in the discussion.

3) Please discuss the relationship between pH changes and eATP-induced defenses.

*Reviewer #2 (Recommendations for the authors):*

The manuscript by Kesten et al. addresses a significant gap in our understanding of plant-pathogen interactions by highlighting the role of fungal pathogen-derived extracellular adenosine (eAdo) in modulating extracellular ATP (eATP) signaling. This novel perspective on the regulation of purinergic signaling in plants has the potential to open up a new field of study. The authors' research demonstrated the important role of eAdo in controlling the eATP/eAdo ratio in the apoplast and its influence on eATP-mediated plant defense responses. The findings suggest that altering this ratio can enhance susceptibility to pathogens, offering valuable insights into the plant's immune response mechanisms. Despite its promising insights, the manuscript leaves certain aspects unresolved, particularly concerning the precise mechanisms of how eAdo is perceived and the exact impact of eATP/eAdo ratio on plant responses to eATP. Further research and exploration are needed for a full understanding of these mechanisms and can be pursued in the future.

The authors have made significant improvements to the original manuscript. I only have a few minor comments that could improve the current version further:

Line 28: The term "secondary messengers" used for eATP and possibly eAdo is not accurate in the context of cell biology. Since eATP initiates a series of downstream signaling events, it should be referred to as a primary messenger. Secondary messengers, like cytosolic ca^2+^, serve to amplify signals generated by the binding of a primary messenger to specific cell surface receptors.

Line 110: I found the term "ATP equilibrium" unclear. It would be helpful if the authors could clarify what it means. I suspect that the authors are referring to a mechanism involving negative feedback to regulate excessive eATP signaling.

Lines 245-246: While references 41 and 42 were cited, their placement in the text appears inappropriate. These papers are still relevant and should be mentioned somewhere in the Discussion section with a more appropriate context.

I'm still finding it challenging to fully grasp how the ATP/Ado ratio and pH alterations might influence a plant's susceptibility. For example, in Figure 1D, an ATP/Ado ratio of 1:2 appears to inhibit eATP-induced resistance. Moreover, at an ATP/Ado ratio of 1:5, eATP-induced pH elevation is suppressed, while a ratio of 1:20 enhances the pH elevation caused by eATP. What might offer a reasonable explanation for the relationship between pH changes and the defense mechanism triggered by eATP?

*Reviewer #3 (Recommendations for the authors):*

I still think that a PCR confirmation of the mutans used (lack of transcript) would strengthen the work.

---

## [Author Response]

[Editors’ note: the authors resubmitted a revised version of the paper for consideration. What follows is the authors’ response to the first round of review.]

Comments to the Authors:We are sorry to say that, after consultation with the reviewers, we have decided that this work will not be considered further at this stage for publication by eLife. The requested experiments would likely exceed a normal time frame for a revision and it is our policy to prevent authors from multiple rounds of revisions with an uncertain outcome. We would, however, be willing to consider a revised manuscript that has fully addressed the main criticisms raised by reviewers.The enhanced calcium signal elicited by by adenosine contradicts the increased susceptibility, and all reviewers believe this is a major unresolved issue. Furthermore, the assumption that Fusarium oxysporum produces adenosine to enhance penetration is not fully supported by the data, as pointed out by reviewer #3. In addition, the use of ent3 nsh3 double mutant is questioned because the mutant has the same amount of adenosine, which weakens the claim.

Many thanks for your time and effort to review our work. We deeply appreciate all comments, suggestions, and criticisms from the Reviewers. We have addressed all of them point-by-point below. In particular, we want to highlight here our replies to the main concerns:

a)Enhanced ca^2+^signal: our data show that Ado only alters the ATP-induced rapid [ca^2+^]_cyt_ transient peak but not the ATP-induced sustained [ca^2+^]_cyt_ elevation. Blume et al. (2022) reported that “Sustained concentrations of [ca^2+^]_cyt_ but not the rapidly induced [ca^2+^]_cyt_ transient peak are required for activation of defense-associated responses”. Therefore, our data indicate that Ado influence on plant-Fo interaction is not [ca^2+^]_cyt_ elevation dependent.b)Secretion of Ado by Fo5176 during root infection: We detected the 5’NT (g8638) in the proteomic secretome of Fo5176-infected roots at 3dpt (Gámez-Arjona et al., 2022)c)Our data show that *ent3nsh3* has a higher Ado/ATP ratio in the media than WT plants, supporting the use of this line. We have modified the figures and text to show these data clearly.

We hope that these and the rest of our arguments and the new version of the manuscript will solve all concerns and our work can be reconsidered for publication by *eLife*.

Reviewer #1 (Recommendations for the authors):The authors identified Ado as an active elicitor produced by F. oxysporum and show that fungal genes, but not plant genes, controlling the production of Ado are specifically induced during infection. Like many elicitors, Ado likely plays a role in virulence, as the addition of Ado can block ATP-induced protection of plants against the fungal pathogen. Similarly, the authors show that the ent3 nsh3 double mutant which has an elevated eAdo/eATP ratio displayed increased susceptibility to the fungus. The authors provide data to show that eAdo can alter eATP signaling in a manner dependent on the ratio. Overall the work presents novel findings that homeostasis of eAdo and eATP plays an important role in plant resistance/susceptibility to the pathogen, which adds a twist to our understanding of the plant immune system. The limitation is with the underlying mechanism of disease susceptibility caused by eAdo/eATP change, as ca^2+^ signaling and apoplast alkalization investigated in this study do not appear to explain the phenotype of plants.

Many thanks for the summary and comments. Please, read below our response to your concerns.

Some of the results observed can be confusing to readers and the authors need to clearly discuss. The alterations of eATP-triggered cytosolic ca^2+^ and apoplastic alkalization by eAdo do not explain the virulence function of eAdo. Ado apparently enhances the ATP-induced ca^2+^ at an eAdo/eATP ratio of 5:1. As ca^2+^ increase is known to be essential for plant immunity, the particular effect of Ado on ca^2+^ described here contradicts our knowledge of ca^2+^ signal in immunity. eAdo blocks eATP-induced alkalization, and the authors claim that blocking eATP-induced pH increases explains the virulence function of eAdo. Although apoplast alkalization is a well-known PTI response, whether and how this impacts disease resistance is not known. In addition, the ent3 nsh3 mutant has a high apoplast pH but is susceptible to F. oxysporum. Thus the authors need to be careful in interpreting the data. It is fine to use ca^2+^ and alkalization as a proxy to show that the eAdo/eATP ratio is important for the alteration of ATP signaling, but at the same time, the authors need to tell readers that these responses do not explain resistance/susceptibility! Thus the authors are strongly encouraged to tone down their statement and address the limitation of their findings.

Many thanks for pointing out these confusions that we address here and in the new version of the manuscript.

Regarding the Ado enhancement of the ATP-induced [ca^2+^]_cyt_ elevation, our data show that Ado only alters the ATP-induced rapid [ca^2+^]_cyt_ transient peak but not the ATP-induced sustained [ca^2+^]_cyt_ elevation. Blume et al. (2022) reported that “Sustained concentrations of [ca^2+^]_cyt_ but not the rapidly induced [ca^2+^]_cyt_ transient peak are required for activation of defense associated responses”. Therefore, our data indicate that Ado influence on plant-Fo interaction is not [ca^2+^]_cyt_ elevation dependent.The impact of apoplastic alkalization in plant-*F. oxysporum* has been shown and discussed in previous works. Based on the current literature, the Ado-induced apoplast alkalization explains the higher susceptibility for two reasons: (a) a fast apoplastic acidification induced by Fo5176 hyphae contact seems to be required for plant defence to this fungus (Kesten et al., 2019) and (b) during Fo colonisation of the plant apoplast, the alkalization of this environment is required for Fo virulence (Masachis et al., 2016).

We have also followed your advice rewording our statements and clarifying the limitations of our findings.

Reviewer #2 (Recommendations for the authors):This work on purinergic signaling in plants, and the influence of a pathogenic fungus on this represents a highly interesting topic. The combination of calcium response quantification with mutants involved in eATP signalling and control of apoplastic adenylate levels allows a deeper insight into this complex process.It is shown that eAdo exerts different responses, it increases calcium responses, and leads to alkalization of the apoplast but counteracts the effect of eATP on root penetration. It is proposed that eAdo activates the plant's immune response, but root growth in the absence or presence of the pathogen is not affected by this metabolite. This is a complex scenario and I have difficulties following the authors' hypothesis. A scheme showing how eAdo interacts with eATP for receptor activation, and how apoplast pH and removal of eAdo is achieved could be helpful in explaining this better.

Thanks for the recommendation of including a scheme that we have followed. We hope that, together with the changes in the text, itallows for a better understanding of our hypothesis.

I miss the verification of the mutants that were reidentified (dorn1, ent3) and the crossed line ent3/nsh3. Were the same T-DNA insertion lines used as in the corresponding original publications? Please give this information in the methods section together with the line identifiers. Also please show a PCR check on cDNA.

Yes, as indicated in the Methods section, we used published mutant lines previously described as KOs. For clarification, we have added these same citations the first time we mentioned the mutants in Results.

Line 72ff: What does it mean, ado is the main component in elicitor mixes.What is an elicitor mix? How many substances were detected in total? Is the method quantitative? Which percentage is ado then?

The elicitor mix is a crude extract made by resuspending and autoclaving Fo5176 mycelia. We did not characterise the total chemical composition of this extract. We fractionated and purified the compounds in the extract using bioassay guided fractionation and only characterised the final pure biologically active component from the mix. Thus, we do not know how many compounds are in the elicitor mix, nor the percentage of Ado in the mix. We have adjusted the text to clarify this information.

Difficult to see the colors in figure 1A, optimize the presentation.

Thanks for the suggestion. We have modified the panel accordingly.

Line 101: Why do the authors write "DORN1 impairment", isn't it a knockout?

Yes, it is a KO. With “impairment”, we meant “damaged function” but we have reworded it to “Lack of DORN1”

Figure 2 D, E use of different symbols in addition to only colors would allow for easier access to the data.

Thanks for the suggestion. We have modified the graphs accordingly. In addition, we have included a graph showing the Ado/ATP ratio (new Figure 2D), which we think helps illustrating the differences on this balance observed among the plant genotypes and treatments. The Ado and ATP graphs are Figure S3.

Reviewer #3 (Recommendations for the authors):The topic of the authors' study has a great impact and provides new insight into the purinergic signal and its homeostasis involved in plant-pathogen interactions.Strengths:Homeostasis of eATP has been studied for root growth, gravitropism, stomata opening/closure, and plant-symbiont interactions, while very limited information on this topic is currently available for plant-pathogen interactions. The research by Kesten et al. suggested an important role of the ATP level in apoplast that was targeted by the fungal pathogen Fusarium oxysporum. The authors proposed that the fungal pathogen releases Ado to regulate the eATP-induced plant defense during infection. There are only a few case studies regarding the Ado function in plants by Roux's group and Mohlmann's group where its mode of action has long been unclear as a negative regulator of the eATP signaling. Thus, the topic in the authors' manuscript has great potential to open up a new field of study in a regulatory mechanism of purinergic signaling in plants.Weaknesses:The discussion in the manuscript was incoherent and scarce, especially regarding their inconsistent results. For example, Ado inhibited the eATP-induced defense, whereas it further enhanced eATP-induced calcium elevation and alkalinization. An experiment was conducted under the assumption of high Ado in a mutant (ent3nsh3), while the actual level of Ado in the mutant was comparable with that in the wild-type plant. The authors need to make convincing explanations.

Many thanks for your comments. We have expanded and modified the discussion of our data in context with the current literature. We paid particular attention on the two main concerns you and the other Reviewers address: (1) “Ado inhibited the eATP-induced defense, whereas it further enhanced eATP-induced calcium elevation and alkalinization” and (2) “An experiment was conducted under the assumption of high Ado in a mutant (ent3nsh3), while the actual level of Ado in the mutant was comparable with that in the wild-type plant”. Please, see our detailed responses below. We do hope that these explanations address your concerns satisfactorily.

I would like to support the study in the manuscript by Kesten et al., given the impact of the topic and the scientific merits of the authors' discovery of the novel function of Ado in plant-pathogen interaction. The results appear to be preliminary and incomplete for publication at the present time. In addition, the discussion in the manuscript was incoherent and scarce.

Thank you for your willingness to support our study. We have modified the text and hope now you find it coherent and complete.

The experiments using the ent3nsh3 mutant were performed under the assumption of the high level of endogenous Ado accumulated in the mutant. If this is the case, the effect of eATP could be nullified without the exogenous application of Ado. Moreover, Figure 2D showed that it was not actually a high level of Ado accumulated in the mutant. The authors are also recognizing that the eAdo level in ent3nsh3 was not remarkably different in comparison to WT (Line 229-231). Unfortunately, all experiments using the mutant are not actually supporting the authors' conclusions in Figure 2-5.

Many thanks for pointing out this discrepancy. Indeed, Daumann et al. (2015) detected increased apoAdo levels in leaves of this mutant compared to WT ones only of soil-grown, but not hydroponically-grown plants. This work shows that higher levels of endogenous Ado in *ent3nsh3* organs cannot be assumed. We have reworded the text to clarify these data and to point out the limitations of the material available.

Our data, indeed, showed no differences between *ent3nsh3* and WT secreted Ado by roots of mock plants grown hydroponically. On the other hand, we show that the apoplastic Ado/ATP ratio is higher in the mutant than in WT. Moreover, upon Fo5176 infection, we detect a significant increase in apoAdo and Ado/ATP ratio in *ent3nsh3* compared to WT roots. These apoplastic purinergic imbalance might explain the different responses of *ent3nsh3* to Fo5176 infection and eAdo treatments, supporting our conclusions in Figures 2-5. We have included an Ado/ATP ratio graph in Figure 2 (Figure 2D) and rewrote the discussion accordingly.

The authors did applaudable work isolating Ado as a potential inhibitor of eATP signaling out of a compound mix secreted from the fungal pathogen. However, validation of the Ado secretion from the fungi was not substantially demonstrated. Data only with gene expression measurement (Figures 1B and 1C) are just suggestive but not conclusive. Does the enzymatic activity of Fo5'NT (or FoENT) always correlate with its transcription? Does the protein level correlate with the transcription level of a fungal enzyme gene? Does an experiment using an isotope directly support the authors' conclusion?

Many thanks for recognizing the chemical effort behind the Ado isolation from the fungal compound mix.

We did aim to fully demonstrate the Ado secretion by Fo5176 by generating Fo5176 mutants unable to secrete Ado (*FoΔE5’NT* and *FoΔENT*). Unfortunately, as we described in the Results section, these efforts failed, most probably due to the vital importance of this process for fungal viability. The upregulation of these fungal genes during root colonisation, while the expression of the plant homologs is not altered, suggest an activation of the Ado secretion in Fo5176 inside the host. These data are supported by the identification of Fo5’NT protein (g8638) in the secretome of Fo5176infected roots of hydroponically-grown Col-0 plants (Gámez-Arjona et al., 2022). We have included this relevant information in the new version and we deeply thank the reviewer for the advice.

It is well known that elevation of cytosolic calcium level is important to induce plant defense response. If eAdo is hypothesized to suppress the eATP-induced defense response, it is expected to reduce the calcium response. However, the authors showed that Ado further enhanced the calcium response, which is not consistent with Ado's inhibitory effect on eATP-induced defense response against the fungal pathogen. It does not make sense at all and the authors did not discuss it in the manuscript. Is it possible to demonstrate that the increased cytosolic calcium level suppresses the defense response and genuinely contributes to the susceptibility against fungal infection?

Our data show that Ado only alters the ATP-induced rapid [ca^2+^]_cyt_ transient peak but not the ATP-induced sustained [ca^2+^]_cyt_ elevation (Figure 3B). Blume et al. (2022) reported that “Sustained concentrations of [ca^2+^]_cyt_ but not the rapidly induced [ca^2+^]_cyt_ transient peak are required for activation of defense-associated responses”. Therefore, our data indicate that Ado’s influence on plant-Fo interaction is not [ca^2+^]_cyt_ elevation dependent. We have included this information in the discussion.

Since eAdo receptors have been reported and well studied in animals, it could be natural to adopt the idea of the eAdo function in plants as proposed by the authors in the Discussion section. However, there are still other possibilities to be tested on how eAdo suppresses eATP signaling. Ado could act as an antagonist and compete with ATP over the DORN1 receptor. It is also possible that eAdo may directly regulate apoplastic enzymes, e.g., ecto-apyrase/E-NTPDase, E5'NT, or another phosphatase, thereby indirectly controlling eATP homeostasis.

Many thanks for these suggestions. We have included these hypotheses in the Discussion of the new version. In addition, we have added the data published by Choi et al. (Science, 2014) showing that there is no competitive inhibition of the DORN1 receptor by Ado; thus, it is not an antagonist of ATP.

I recommend changing the title since "Vascular fungi" provides us the impression that the authors study the eATP signaling in the plant vascular tissue. "Ascomycete fungus" or "Soilborne fungus" would be appropriate for this study given their pathogenicity.

Thank you for the recommendation. We have changed the title.

Line 58: The role of eATP against "different pathogens" was summarized in Table 1 of Jewell et al. https://doi.org/10.1093/plphys/kiac393, which could be an appropriate paper to cite.

We have cited this work in the new version.

Since this manuscript is about the role of eATP as a DAMP against the soilborne pathogen, it would be fair to compare the authors' data with the previous report where the soilborne pathogen Rhizoctonia (Kumar et al. https://doi.org/10.3389/fpls.2020.572920) was used and discuss further.

Our work does not focus on eATP as a DAMP against Fo5176, but we certainly show this role for the first time. Thus, we agree on referring to the mentioned work of P2K1/DORN1 function on plant resistance against another soilborne fungal pathogen.

The term "elicitor" is about a compound that induces (elicits) plant defense. Toxin or effector is a compound that manipulates plant immune systems to help infection of symbionts and pathogens. The term should be distinguished carefully. In addition, I do not agree with defining "eAdo as a MAMP" on lines 203-4.

Thank you for this comment. We agree on the need to clarify these terms. “Elicitor” is “a substance that induces a reaction from an organism”. As such, it is not only related to defence responses. Thus we refer to the fungal compound mix as “elicitor mix” because it does induce a reaction from the plant. Indeed, based on our data, the Ado secreted by Fo5176 inside the root can be considered a fungal “effector”. Thanks for the suggestion. For sure, eAdo is not a MAMP and we apologise for this error that we have corrected.

Since the study in this manuscript is about an apoplastic effector (toxin?) compound by fungal pathogens, the following 3 literature should be introduced together as previous precedents: Masachis et al. https://doi.org/10.1038/nmicrobiol.2016.43; Wei et al. https://doi.org/10.1038/s41467-022-29788-2; Xia et al. https://doi.org/10.1073/pnas.2012149117.

Many thanks for pointing out these works that we have included as precedents in the Discussion of the new version.

Since it is about [ATP] vs. [ATP + Ado], the statistic analysis should be performed between them for Fig1D and Figure 1-supl2.

We did include these statistical analysis in the Suppl Tables (tables S2A and S2C, respectively). As this way of presenting them does not seem to be informative enough, we have also added them in the graphs as blue asterisks.

Not everyone is an expert in the experiment of vascular penetration by fungal hyphae. It would be helpful for the broad audience if the authors kindly show a few representative pictures or diagrams to explain the method.

We did explain the method in detail, with pictures and diagrams, in two previous works (Kesten et al., 2019; Huerta et al., 2020). To avoid redundancy and information overload, we decided just to cite those references. We still think this is the most appropriate option but are happy to include more information about the vascular penetration method, if found necessary.

Why the authors used non-buffered media? ATP is a strong anion and it might have a significant impact on pH in the media. Root growth is known to be sensitive to pH. Maybe the Ado effect was masked by the effect of the pH lowering by the ATP addition. What would be the result if the experiment in Figure 2C is performed by preventing pH changing with a buffer to observe a true effect of ATP as a signal?

As previously reported by Machasis et al. (2016) and ourselves, the dynamic change of the apoplastic pH is essential during Fo infection. A buffered media alters the plant-microbe interaction. Preventing pH changing will affect many molecular aspects of the infection and, thus, will not be informative. Indeed, ATP in pure water results in a mildly acidic pH around 3.5 because its phosphate groups are weakly acidic with pK values in the physiological range. However, this is not the case in our media whose pH does not change after adding ATP or Ado concentrations used throughout the study.

Line 137-8: I do not agree that the response to 10 μm ATP is enough to say "more sensitive than previous report". A previous report showed that even 3 μm eATP provided enough consistent data (Demidchik et al. https://doi.org/10.1104/pp.103.024091).

Many thanks for mentioning this work. We have corrected the text accordingly

Line 161: It is inappropriate to mention "up to 100 uM" without such data.

Thank you for pointing out this typo, that we have corrected to “up to 50 uM”.

Line 166: It is inappropriate to mention "high eAdo/eATP ration in ent3nsh3 apoplast without such data.

We have included a new graph to illustrate this ratio that could be inferred from panels 2D-E from our previous version.

Line 176: How many genes did the authors test for "various" defense-related genes? Can it be specified?

We tested four defense-related genes: *DORN1*, *WRKY45, WRKY53,* and *At1g51890*, which we have indicated in the new version.

Line 196: "just" 10 μm ATP? What are the authors emphasizing?

We have corrected it in light of Demidchik et al., 2003.

Line 199 and 200: My understanding is that the inhibition effect of Ado may be not due to a chemical reaction. So, "Ado itself was inactive" should be rephrased.

We have rephrased to “Ado alone did not induce any plant response different from the control treatment “. We do hope that now our conclusion is clear.

Line 205-7: The sentence does not make sense. How can eAdo block plant defense by blocking eATP-induced alkalinization even though alkalinization is for fungal pathogenesis? Instead, Figure 4B, where Ado further enhanced alkalinization, seems reasonably consistent with the previous case in ref 23.

Thank you for pointing this out. We modified the sentence.

Line 220: Since ATP is a strong anion, the acidification could be due to its chemical feature since no alkalinization in dorn1 was induced, as an alternative.

Absolutely, many thanks for pointing this out. As we mentioned above, although ATP in pure water results in a mildly acidic pH around 3.5, it does not alter the pH of the media we use, nor Ado. Thus, we have not included this possibility in the text.

Line 221: The authors pointed out that P2K2 is involved in eATP-induced apoplastic acidification in Figures4E and 4F. Please explain and discuss further in detail.

We extended the part about P2K2 in the current discussion.

Line 221-2: I had a tough time to understand how GPA is involved in the acidification in dorn1. Please elaborate more.

Hao et al. (2012) showed that “In null mutants of the heterotrimeric G protein α subunit, ATP-promoted stomatal opening, cytoplasmic ROS generation, ca^2+^ influx, and H^+^ efflux were all suppressed”. This indicates that GPA is required for apoplast acidification in response to ATP under these conditions. Thus, it is likely that GPA also plays a role in acidification of the apoplast in another context.

Line 247: I am not sure that it is a concentration-dependent manner since 10 μm and 200 μm of Ado evoked the opposite responses (Figure 4A and 4B). Please rephrase in an appropriate manner.

We deleted in a ‘concentration-dependent manner` and left the sentence more broadly defined.

Line 249-51. I do not understand the sentence. Did the authors mean ENT3 is directly regulating (or transporting) protons since the authors speculated that phosphorylation of ENT3 by eATP is required for Ado-elicited pH changes?

There is conflicting data in the literature whether ENT3 is a proton symporter (references *10*, *35*–*37* in the manuscript). Following these results, we, thus, hypothesised that ENT3 might indeed need to be phosphorylated by e.g. a component of the DORN1-mediated signalling cascade as reported in mammalian systems.

The following sentence on lines 251-3 was more confusing. Do the authors hypothesize ENT3 as an H^+^ transporter?

We do not hypothesise it to be a transporter but there is conflicting data about if ENT3 is a proton symporter (not transporter) while transporting Ado in the literature (references *10*, *35*–*37* in the manuscript).

We have added more information in the text to explain this hypothesis.

Line 252-55: The authors speculate that there are H^+^-ATPases for ATP/Ado-induced pH changes. Indeed, ATP-induced calcium response and apoplastic pH (also cytosolic pH) are reported to be associated based on the data in Behera et al. https://doi.org/10.1105/tpc.18.00655 and in Figure 5D in Haruta et al. https://doi.org/10.1104/pp.111.189167. It would be nice to discuss this further by citing these publications.

We included both publications now.

I do not understand the logic of the authors' conclusion on lines 262-3. The authors speculated that there is a potential Ado receptor because eATP signaling is required for the inhibition effect by Ado.

We modified the sentence and hope it is more understandable now.

References

Blume B, Nürnberger T, Nass N, Scheel D. Receptor-mediated increase in cytoplasmic free calcium required for activation of pathogen defense in parsley. Plant Cell. 2000 Aug;12(8):1425-40. doi: 10.1105/tpc.12.8.1425. PMID: 10948260; PMCID: PMC149113.

Gámez-Arjona FM, Vitale S, Voxeur A, Dora S, Müller S, Sancho-Andrés G, Montesinos JC, Di Pietro A, Sánchez-Rodríguez C. Impairment of the cellulose degradation machinery enhances *Fusarium oxysporum* virulence but limits its reproductive fitness. Sci Adv. 2022 Apr 22;8(16):eabl9734. doi: 10.1126/sciadv.abl9734. Epub 2022 Apr 20. PMID: 35442735; PMCID: PMC9020665.

Huerta AI, Kesten C, Menna AL, Sancho-Andrés G, Sanchez-Rodriguez C. In-Plate Quantitative Characterization of *Arabidopsis thaliana* Susceptibility to the Fungal Vascular Pathogen Fusarium oxysporum. Curr Protoc Plant Biol. 2020 Sep;5(3):e20113. doi: 10.1002/cppb.20113. PMID: 32598078. Hao LH, Wang WX, Chen C, Wang YF, Liu T, X. L X, Z.-L. Shang ZL, Extracellular ATP promotes stomatal opening of *Arabidopsis thaliana* through heterotrimeric G protein α subunit and reactive oxygen species. *Mol. Plant*. **5**, 852–864 (2012).

Kesten C, Gámez-Arjona FM, Menna A, Scholl S, Dora S, Huerta AI, Huang HY, Tintor N, Kinoshita T, Rep M, Krebs M, Schumacher K, Sánchez-Rodríguez C. Pathogen-induced pH changes regulate the growth-defense balance in plants. EMBO J. 2019 Dec 16;38(24):e101822. doi: 10.15252/embj.2019101822. Epub 2019 Nov 18. PMID: 31736111; PMCID: PMC6912046.

Masachis S, Segorbe D, Turrà D, Leon-Ruiz M, Fürst U, El Ghalid M, Leonard G, López-Berges MS, Richards TA, Felix G, Di Pietro A. A fungal pathogen secretes plant alkalinizing peptides to increase infection. Nat Microbiol. 2016 Apr 11;1(6):16043. doi: 10.1038/nmicrobiol.2016.43. PMID: 27572834.

[Editors’ note: what follows is the authors’ response to the second round of review.]

The reviewers have discussed their reviews with one another, and the Reviewing Editor has drafted this to help you prepare a revised submission.Essential revisions:1) Please correct/clarify the terminology pointed out by reviewer #2.2) Cite references 41 and 42 in the discussion.3) Please discuss the relationship between pH changes and eATP-induced defenses.

We have made all these revisions in the text.

Reviewer #2 (Recommendations for the authors):The manuscript by Kesten et al. addresses a significant gap in our understanding of plant-pathogen interactions by highlighting the role of fungal pathogen-derived extracellular adenosine (eAdo) in modulating extracellular ATP (eATP) signaling. This novel perspective on the regulation of purinergic signaling in plants has the potential to open up a new field of study. The authors' research demonstrated the important role of eAdo in controlling the eATP/eAdo ratio in the apoplast and its influence on eATP-mediated plant defense responses. The findings suggest that altering this ratio can enhance susceptibility to pathogens, offering valuable insights into the plant's immune response mechanisms. Despite its promising insights, the manuscript leaves certain aspects unresolved, particularly concerning the precise mechanisms of how eAdo is perceived and the exact impact of eATP/eAdo ratio on plant responses to eATP. Further research and exploration are needed for a full understanding of these mechanisms and can be pursued in the future.The authors have made significant improvements to the original manuscript. I only have a few minor comments that could improve the current version further:Line 28: The term "secondary messengers" used for eATP and possibly eAdo is not accurate in the context of cell biology. Since eATP initiates a series of downstream signaling events, it should be referred to as a primary messenger. Secondary messengers, like cytosolic ca^2+^, serve to amplify signals generated by the binding of a primary messenger to specific cell surface receptors.

Thanks for this correction, we have modified the text accordingly.

Line 110: I found the term "ATP equilibrium" unclear. It would be helpful if the authors could clarify what it means. I suspect that the authors are referring to a mechanism involving negative feedback to regulate excessive eATP signaling.

Yes, you are right, thanks. We have modified the text to “eATP signaling regulation”.

Lines 245-246: While references 41 and 42 were cited, their placement in the text appears inappropriate. These papers are still relevant and should be mentioned somewhere in the Discussion section with a more appropriate context.

We have relocated them in the discussion.

I'm still finding it challenging to fully grasp how the ATP/Ado ratio and pH alterations might influence a plant's susceptibility. For example, in Figure 1D, an ATP/Ado ratio of 1:2 appears to inhibit eATP-induced resistance. Moreover, at an ATP/Ado ratio of 1:5, eATP-induced pH elevation is suppressed, while a ratio of 1:20 enhances the pH elevation caused by eATP. What might offer a reasonable explanation for the relationship between pH changes and the defense mechanism triggered by eATP?

Our short-term response data suggest that the rapid apoplastic alkalization generated by eATP is not a main contributor to plant defence against Fo5176, as (1) eAdo boosts this plant response potentially leading to increased fungal virulence (Figure 1D and 4B), and (2) eATP also generates a pH_apo_ peak in the Fo5172 susceptible mutant ent3nsh3, similar to WT plants. It was previously reported that eATP induces defence-related gene expression independently of its effect on media pH (Jewell et al., 2019). However, more research is needed to clarify the relationship between pH changes and the defence mechanism triggered by eATP, and MAMPs/DAMPs/elicitors in general, in the process of an infection. We have included these comments in the discussion.

About the effect of the different externally altered ATP/Ado ratio in plant responses, we think it is important to keep in mind that we can not numerically compare the results from plate infection assays with that we observe in short response analysis of pH_apo_ changes. In the plate infection assays, Ado and ATP were dissolved in the media where the plants grew for 7 days and Fo5176 increased the apoplastic Ado level during the infection (see Figure 2D WT mock vs infected, among other data). The pH_apo_ rapid changes induced by different ATP/Ado were tested in 10 min assays, in a concrete area of the root, and without additional effects caused by a pathogen. As the media surrounding plants colonised by Fo strongly alkalizes during the infection (Masachis et al., 2016; Kesten et al., 2019), we hypothesise that the ATP/Ado levels reach the ratio above which there is a boost of ATP-induced alkalization in our plate-infection assays. Thanks to your question we have realised that the sentence in the Discussion “Indeed, a minimum amount of eAdo is required to block the ATP-induced plant resistance to Fo5176 (Figure 1D and S2)” is not correct and have decided to remove it.

Reviewer #3 (Recommendations for the authors):I still think that a PCR confirmation of the mutans used (lack of transcript) would strengthen the work.

All mutants have been characterised before, so we do not think a reconfirmation to be necessary. As this is also not included in the Essential Revisions of the Editors, we kindly decline to do this assay.

References

J.B. Jewell, J.M. Sowders, R. He, M.A. Willis, D.R. Gang, K. Tanak. Extracellular ATP Shapes a Defense-Related Transcriptome Both Independently and along with Other Defense Signaling Pathways. Plant Physiol. 2019 Mar;179(3):1144-1158. doi: 10.1104/pp.18.01301. Epub 2019 Jan 10. PMID: 30630869; PMCID: PMC6393801.

S. Masachis, D. Segorbe, D. Turra, M. Leon-Ruiz, U. Furst, M. El Ghalid, G. Leonard, M. S. Lopez-Berges, T. A. Richards, G. Felix, A. Di Pietro, A fungal pathogen secretes plant alkalinizing peptides to increase infection. Nat Microbiol. **1**, 16043 (2016).

C. Kesten, F. M. Gámez‐Arjona, A. Menna, S. Scholl, S. Dora, A. I. Huerta, H. Huang, N. Tintor, T. Kinoshita, M. Rep, M. Krebs, K. Schumacher, C. Sánchez‐Rodríguez, Pathogen‐induced pH changes regulate the growth‐defense balance in plants. The EMBO Journal. 38 (2019), doi:10.15252/embj.2019101822.